# A very limited role of tropospheric chlorine as a sink of the greenhouse gas methane

Sergey Gromov[1,*], Carl A. M. Brenninkmeijer[1] and Patrick Jöckel[2]

[1] Max Planck Institute for Chemistry, Atmospheric Chemistry Department, Mainz, Germany

[2] Deutsches Zentrum für Luft- und Raumfahrt (DLR), Institut für Physik der Atmosphäre, Oberpfaffenhofen, Weßling, Germany

[*] Also at Institute of Global Climate and Ecology Roshydromet & RAS (IGCE), Moscow, Russia

*Correspondence to*: Sergey Gromov (sergey.gromov@mpic.de)

## Abstract

Unexpectedly large seasonal phase differences between $CH_4$ concentration and its $^{13}C/^{12}C$ isotopic ratio and their inter-annual variations observed in southern hemispheric time series have been attributed to the $Cl+CH_4$ reaction, in which $^{13}CH_4$ is discriminated strongly compared to $OH+CH_4$, and have provided the only and indirect evidence of a hemispheric-scale presence of oxidative cycle-relevant quantities of tropospheric atomic Cl. Our analysis of concurrent New Zealand and Antarctic time series of $CH_4$ and CO mixing and isotope ratios shows that a corresponding $^{13}C/^{12}C$ variability is absent in CO. Using the AC-GCM EMAC model and isotopic mass balancing for comparing the periods of presumably high and low Cl, it is shown that variations in extra-tropical Southern Hemisphere Cl can not have exceeded $0.9 \times 10^3$ atoms $cm^{-3}$. It is demonstrated that the $^{13}C/^{12}C$ ratio of CO is a sensitive indicator for the isotopic composition of reacted $CH_4$ and therefore for its sources. Despite ambiguities about the yield of CO from $CH_4$ oxidation, with this yield being an important factor in the budget of CO, and uncertainties about the isotopic composition of sources of CO, in particular biomass burning, the contribution of Cl to the removal of $CH_4$ in the troposphere is probably much lower than currently assumed.

## 1 Introduction

[1] Compared to the troposphere's main oxidant OH (hydroxyl radical), the role of Cl (atomic chlorine) for $CH_4$ is small. A recently published detailed model-based estimate attributes ~2.6% of methane's photochemical tropospheric loss to Cl (Hossaini *et al.*, 2016). Because this loss constitutes only a small term in the methane budget, it might be deemed not to be relevant. Nevertheless, growing spatial and temporal coverage in $CH_4$ observational data allows for top-down estimates of changes in the source-sink budget to the order of ~1%. Moreover, considering that the photochemical sink is the dominant and best-known term in the global methane budget, it makes sense to improve our calculations. The grateful aspect of this endeavour clearly is that one does not need an accurate estimate of Cl as a global tropospheric sink of $CH_4$ as such. It would already be helpful to have independent estimates of the upper limit for this interesting sink of $CH_4$, whose rise in the Anthropocene thus far has contributed $^1/_5$ to global warming.

[2] Irrespective of the implications for the $CH_4$ budget, it stands to reason to fully understand tropospheric Cl and its chemistry in different air masses, from marine boundary layer air to strongly polluted air masses and several studies address these complex processes. It is also clear, that the budget of a species as fickle as atomic chlorine is hard to determine in general terms (which forms a less grateful aspect of "assessing chlorine"). Nevertheless, a new effort – in assessing chlorine's role on a larger than regional scale, on the basis of trace gas measurements, may be useful.

[3] Even more so than for OH, estimates of the abundance of Cl atoms are chiefly based on indirect evidence. Direct measurements of OH concentrations ([OH]) being difficult and rare, for [Cl] this is even much more so. Therefore, the method (by choice or opportunity) is indirect. Not only are indirect measurement easiers, the use of trace gases that react with OH and Cl also has the advantage that space- and time-averaged estimates are obtainable. In this case, one can select for instance 2 hydrocarbons one of which has a comparatively high reactivity to Cl. The change in ratio between the two hydrocarbon concentrations gives information on [Cl] relative to [OH].

[4] Using stable isotope ratio information offers another such indirect method. The intrinsic advantage here is that one can use a single trace gas, a single hydrocarbon, or even the much studied greenhouse gas $CH_4$ itself. Although the rate coefficient for the reaction of OH with $^{12}CH_4$ is only ~4‰ faster than that with $^{13}CH_4$ (Saueressig *et al.*, 2001), for $Cl+CH_4$ the difference is much larger (Saueressig *et al.*, 1995; Crowley *et al.*, 1999), *viz.* (63–75)‰ (at the range of tropospheric temperatures). Broadly speaking, the presence of $^{13}C$ enriched $CH_4$ points to reaction with Cl. If this were not enough, one could measure the D/H ratio of $CH_4$ and obtain additional valuable information because of the large isotope fractionation (KIE, Kinetic Isotope Effect, formerly and still expressed using the kinetic fractionation constant $\varepsilon = \alpha - 1$) and the differences between the KIEs for $^{13}C$ and D. A recent paper (Whitehill *et al.*, 2017) reports changes in the clumped isotopic composition of $CH_4$ in reaction with Cl based on laboratory experiments, raising hope that clumped isotope measurements (which are very difficult) may in an additional

way assist to further assess the role of Cl in the oxidation of $CH_4$ in the atmosphere.

[5] An advantage is that the "stable isotope method" in principle removes the uncertainty about variability induced by having to use two different trace gas species, each of which may have an independent, variable source. Routinely overlooked is another (principle) advantage of stable isotope analysis offered in the case of atmospheric $CH_4 \rightarrow CO$ conversion, namely measurement of the isotopic composition of the reaction product CO. Even though variations in [CO] may not be resolvable due to the large spatio-temporal variability of its sources and sink, its $^{13}C/^{12}C$ ratio may well tell a clearer story. This is the added advantage of the stable isotope method (we note that the lifetime of $^{14}C$ is sufficiently long to render much of what is stated to also apply to this well-known radioisotope, but there are complications on which we cannot dwell here).

[6] In this way the presence of Cl during Antarctic ozone hole conditions could be inferred in an independent fashion (Brenninkmeijer *et al.*, 1996). Not only became the $CH_4$ inventory slightly enriched in $^{13}C$ due to the large KIE in $Cl+CH_4$, the CO ensuing from $CH_4$ resulted in strong depletions in $^{13}C$ of background CO. There are at least three reasons for the strong isotope depletion. Firstly, CO concentrations are low in the stratosphere and the in situ produced CO had a large impact. Secondly, the $^{13}C$ content of $CH_4$ is characteristically low due to its chiefly bacterial origin. Thirdly, and this is an important point mentioned above, the $^{13}C$ KIE for $Cl+CH_4$ happens to be very large. The combination of these effects renders the stable isotope analysis of CO a sensitive indicator. Dealing with tropospheric Cl, the same principle has been applied during springtime tropospheric ozone depletion events in the Arctic. Short-term bursts of free Cl could be inferred from concomitant decreases in $\delta^{13}C(CO)$ within a per mil[1] range (Röckmann *et al.*, 1999).

[7] We record that there also is a removal of CO by reaction with Cl atoms with the rate constant being typically six times smaller than that of CO+OH. Given this very low rate coefficient and the low Cl/OH ratio, only an extremely large KIE in CO+Cl reaction could impact significantly on $\delta^{13}C$ of the CO inventory. In contrast, the rate constant for $CH_4+Cl$ is typically 20 times larger than that for $CH_4+OH$. Cl is not expected to play a significant role in atmospheric CO removal, except possibly at polar sunrise (Hewitt *et al.*, 1996) and in some stratospheric chemistry analyses (see, *e.g.*, Müller *et al.* (1996), Sander *et al.* (2011b)). None of a few of papers on tropospheric CO thus mentions Cl as a sink for CO because of its negligible share; fortunately, because the reaction product is not so nice.

[8] In this brief account we cannot do justice to all tropospheric Cl related papers in the literature and we refer to the recent model based paper by Hossaini *et al.* (2016) and references therein. In comparison with OH, which is recycled in about two of three reactions in the troposphere (Lelieveld *et al.*, 2016), the role of recycling of Cl is lower and not known well. The presence of Cl in the marine boundary layer has been inferred using hydrocarbon measurements (early reference Parrish *et al.*, 1993) and likewise during polar sunrise (Jobson *et al.*, 1994), $Cl_2$ has been measured in situ in coastal air (Spicer *et al.*, 1998) and in the Arctic (Liao *et al.*, 2014). $ClNO_2$, which is an important precursor, has been measured (Osthoff *et al.*, 2008 and Thornton *et al.*, 2010), also by Young *et al.* (2012), who however found no Cl fingerprint in hydrocarbon ratios.

[9] Recently, Baker *et al.* (2016) inferred the presence of Cl in pollution outflow from continental Asia using hydrocarbon measurements on air samples collected at cruise altitude by the CARIBIC Lufthansa Airbus aircraft observatory. Before that, Baker *et al.* (2011) had likewise inferred Cl being formed in an emission plume of the Eyjafjallajökull volcano probed by the same CARIBIC A340 aircraft. All these and other publications discuss the presence of Cl in a variety of tropospheric environments wrestling with the complexity of its chemistry and paucity of experimental data.

[10] Additional importance of revisiting the role of Cl radicals in the present atmosphere actually surfaces in the reconstruction and understanding of the budget of $CH_4$ in the past. Changes in the tropospheric burden of $CH_4$ that occurred in the past (last glacial maximum – present) are due to changes in $CH_4$ sources and to a minor degree to changes in OH chemistry (Levine *et al.*, 2011b). One would *a priori* expect $\delta^{13}C(CH_4)$ to provide additional information on source changes, as it did for immediate past changes (Schaefer *et al.*, 2016), were it not that large changes in Cl abundance may well have affected the $\delta^{13}C(CH_4)$ record (Levine *et al.*, 2011a). If this is the case indeed, changes in Cl abundance in the past may have not affected the $CH_4$ budget itself significantly, but may have invalidated to a certain degree the $\delta^{13}C(CH_4)$ isotope method for determining changes in sources (biogenic *vs.* biomass burning).

[11] We turn our attention to a paradox concerning today's tropospheric Cl, namely: If the presence of tropospheric Cl could be inferred from $^{13}C$ isotope enrichment in $CH_4$, why is this effect not visible as concurrent isotope depletion in CO? Or, more explicitly stated, if the $\delta^{13}C(CO)$ isotope method for Cl detection works well for the austral polar stratosphere in spring (Brenninkmeijer *et al.*, 1996) and for the polar sunrise in the Arctic (Röckmann *et al.*, 1999), why not so for the troposphere, or does it? Is a clear negative signal in $\delta^{13}C(CO)$ absent indeed, and if so, does this absence allow us to cap estimates of tropospheric Cl levels?

## 2 Data analysis

### 2.1 Chlorine in the Southern Hemisphere

[12] Because the budgets of $CH_4$ and CO in the Southern Hemisphere (SH) are less complicated than in the Northern Hemisphere, as is witnessed by their compact regular seasonal cycles at remote observatories[2], and because long records of CO and

---

1  Hereinafter we report the $^{13}C/^{12}C$ ratio as per mil delta values. The $\delta^{13}C$ is defined as $\delta^{13}C = (R/R_{st} - 1)$, where $R$ and $R_{st}$ denote the sample and standard $^{13}C/^{12}C$ ratios. We use the V-PDB scale with $R_{st} = 11237.2 \times 10^{-6}$ (Craig, 1957) throughout this paper (for details on choosing this value see Gromov *et al.*, 2017, Appendix A).

2  See, *e.g.*, the synthesis of the CO and $CH_4$ observational data at https://www.esrl.noaa.gov/gmd/ccgg/gallery/figures/ and refs. provided therein (last access: December 2017).

CH$_4$ including isotopic data are available, we focus on the Southern Hemisphere. In the SH evidently the emphasis is on Cl generated in the marine boundary layer (MBL).

[13] We first revisit the information on Cl based on $\delta^{13}$C measurements of CH$_4$. Initially, mixing ratio and $\delta^{13}$C(CH$_4$) values for shipboard collected air samples in the Pacific pointed to a large apparent sink isotope fractionation ("apparent" KIE) of (12−15)‰ – well in excess of the aforementioned 4‰ from OH+CH$_4$ – which led to the conjecture that a fraction of CH$_4$ is removed in the MBL by Cl atoms which discriminate strongly against $^{13}$CH$_4$ (Lowe *et al.*, 1999; Allan *et al.*, 2001). Following several publications exploring this effect, Allan *et al.* (2007) (hereinafter referred to as A07) using global modelling and observational data from the extratropical Southern Hemisphere (ETSH), confirmed a large apparent KIE and could estimate a global marine boundary layer based Cl sink for CH$_4$ averaging at 25 Tg(CH$_4$) yr$^{-1}$.

[14] Given this number, a first order estimate of the accompanying response of $\delta^{13}$C of CO to the production of CO from Cl+CH$_4$ can be made. Assuming a 100% yield of CO from OH+CH$_4$ (and likewise Cl+CH$_4$), the 25 Tg yr$^{-1}$ CH$_4$ sink corresponds to a Cl based annual CO production of 44 Tg yr$^{-1}$, which is ~1.8% of the total CO budget. By using a $\delta^{13}$C value of CO of −28‰ (annual tropospheric average), that of CH$_4$ of −48‰ and a KIE of 70‰, (Cl+CH$_4$) causes a negative shift in $\delta^{13}$C(CO) of about 1.6‰. Considering that the lifetime of CO is much shorter than that of CH$_4$ and that Cl is concentrated in the MBL, the local/seasonal effect on $\delta^{13}$C(CO) would be even larger.

[15] Unfortunately, a negative shift in $\delta^{13}$C(CO) is upfront unwelcome in attempts to close the SH CO budget using $\delta^{13}$C. As Manning *et al.* (1997) have pointed out, budget closure is only possible when the yield of CO from CH$_4$+OH (denoted hereinafter $\lambda$) is assumed to be merely about 0.7. In other words, even without incorporating the formation of CO from Cl+CH$_4$, the CH$_4$-derived $^{13}$C-depleted fraction of CO (which is high in the ETSH at above 40%) appeared to be too dominant and had to be reduced by assuming lower yields of CO from CH$_4$. Soon thereafter also Bergamaschi *et al.* (2000) encountered this problem in a 3D inverse modelling study using the isotopic composition of CO and could best reconcile data and model by reducing $\lambda$ to about 0.86. They do mention that incorporating CO from Cl+CH$_4$ would require $\lambda$ values as low as 0.71. Also Platt *et al.* (2004) who discuss mechanisms for the production of Cl in the marine boundary layer allude to the necessity to have to reduce the assumed CO yield of OH+CH$_4$.

[16] One difficult feature of the $\delta^{13}$C(CH$_4$)-based Cl estimate was a large inter-annual variability that could not be explained. A07 identified two periods of different Cl abundance in the ETSH, namely 1994–1996 with MBL values of 28×10$^3$ atoms cm$^{-3}$ (high-Cl period, "HC") and 1998–2000 with much lower values, *viz.* 9×10$^3$ atoms cm$^{-3}$ (low-Cl period, "LC"). The nearly threefold drop in the resulting Cl+CH$_4$ sink rate (37 to 13 Tg(CH$_4$) yr$^{-1}$, or 6.4% to 2.2% of the total, respectively) inferred from $\delta^{13}$C(CH$_4$) for the two periods is not discernible in the simultaneous $\delta^{13}$C(CO) record (see Sect.2.2).

[17] Later, Lassey *et al.* (2011) investigated the apparent KIE in detail and found that it can differ markedly from both the seasonal and mass-balanced KIEs. In other words, the apparent KIE derived from the seasonal changes in [CH$_4$] and $\delta^{13}$C(CH$_4$) value appeared not to properly represent the respective effects of the two KIEs. The implication is that the inferred very large range of [Cl] may be in error, and the absence of a corresponding signal in $\delta^{13}$C(CO) is in that respect an experimental confirmation. Below we will go into detail.

## 2.2 Observations in the ETSH

[18] We scrutinise the mixing and $^{13}$C/$^{12}$C ratios of CH$_4$ and CO in the MBL air at Baring Head, New Zealand (41.41°S, 174.87°E, 85 m a.s.l., denoted hereinafter "BHD") and at Scott Base, Antarctica (77.80°S, 166.67°E, 184 m a.s.l., denoted "SCB")[3] provided by the National Institute of Water and Atmospheric Research (NIWA, 2010). Examined in the A07 study on CH$_4$, these data are the result of laboratory analyses of large air samples collected on a monthly to weekly basis. The collection strategy (using wind direction, CO$_2$ mixing ratio temporal stability and back-trajectory analysis) allows selecting air masses that represent background ETSH air. Established over two decades, these time series confer the longest continuous records of $^{13}$CH$_4$ and $^{13}$CO observations to date. The reported overall uncertainties of the CH$_4$ mixing ratio and $\delta^{13}$C do not exceed ±0.3% (about ±5 nmol/mol) and ±0.05‰ (Lowe *et al.*, 1991). For CO, the respective uncertainties are ±4%/±0.2‰ (prior to 1994, Brenninkmeijer, 1993) and ±7%/±0.8‰ (since 1994, NIWA, 2010). The CO records from BHD/SCB exhibit small variations in annual (minimum-to-maximum) span and no significant long-term trend in both mixing and isotope ratios throughout 1990−2005 (see Gromov, 2013, Sect. 4.1.1). In contrast to this, the concomitant [CH$_4$] values have increased on average by about 5% within the same period, which is consistent with other observational records (Lassey *et al.*, 2010). It can be concluded, that such augmentation of atmospheric burden of the major (and largely depleted in $^{13}$C) in-situ sources of CO remains statistically indiscernible in the ETSH $\delta^{13}$C(CO) record, because of more perceptible variations caused by changes in sink and/or the other (foremost biomass burning) sources of CO.

[19] We subsequently regard the statistics of the two subsets of observational data falling into the HC and LC periods, as shown in Fig. 1. For testing the robustness of our comparison against the timing of the air sampling, we "bootstrap" the data by selecting only the pairs of CH$_4$/CO samples collected within one-week windows (shown with solid boxes in Fig. 1). This operation has virtually no effect on CO distributions, as its statistic is smaller (total of 116 and 88 samples at BHD and SCB, respectively) and controls the sub-sampling of the datasets. For CH$_4$, also no effect is noted, with an exception of significant (*i.e.* exceeding measurement uncertainty) changes to the "bootstrapped" median CH$_4$ mixing ratio at BHD, which is some 6 nmol/mol lower during

---

3   Sample collection takes place at the designated clean air site Arrival Heights; some of the NIWA datasets use the abbreviation "AHT" for this site.

the HC. Such is an indication that the CO sampling times are likely more representative for background air. Overall, we conclude that the $CH_4$ and CO datasets reflect variations in the composition of the same background air. Contrary to $CH_4$, there is no perceptible reduction in seasonal variations of mixing and isotope ratios of CO at SCB throughout the HC period.

[20] To determine the significance of observed changes in CO using sufficient statistics, we derive quasi-annual averages (QAA) of CO mixing/isotope ratio averages representing the HC, LC and long-term periods (all data and from 1994 onwards). For the correct temporal weighting of the samples, we first calculated quasi-monthly averages and their variances, which then equally contributed to the QAA. Table 1 lists the results along with the number of samples used in the calculation. Note that there are about twice as many outliers[4] in the entire BHD record (3.8%) compared to that for SCB (2.2%), which suggests that

the estimated difference between the HC and LC averages (HC−LC, denoted $\Delta$) is probably more influenced by regional sources at BHD. Except for $\delta^{13}C(CO)$ at SCB (with considerable significance of $\Delta$ being negative, $p$-value of 0.79), we conclude that all CO QAAs emerge as statistically indistinguishable, also when compared to the long-term averages. For CO mixing ratios, the Cl-driven difference should amount up to 1.2 nmol/mol (conservatively assuming up to 50% of CO derived from $CH_4$ oxidation changed by 4.2%), which is 2.5–3 times smaller than the errors in $\Delta$. At both stations, the $\Delta$ values indicate changes to the atmospheric reservoir involving $^{13}C$-depleted CO, however in opposite directions (*i.e.* a removal at BHD – which contradicts A07 – and an addition at SCB). It is important to note that the CO+OH sink alters atmospheric CO in a similar fashion (*i.e.*, the remaining CO burden becomes enriched in $^{13}C$).

---

[4] We follow the conventions from Natrella (2003) for identifying statistically significant outliers in the datasets. Samples with mixing ratios falling outside inner and outer statistical fences of ±1.5 and ±3 interquartile ranges (IQR) about the median are considered mild and extreme outliers, respectively.

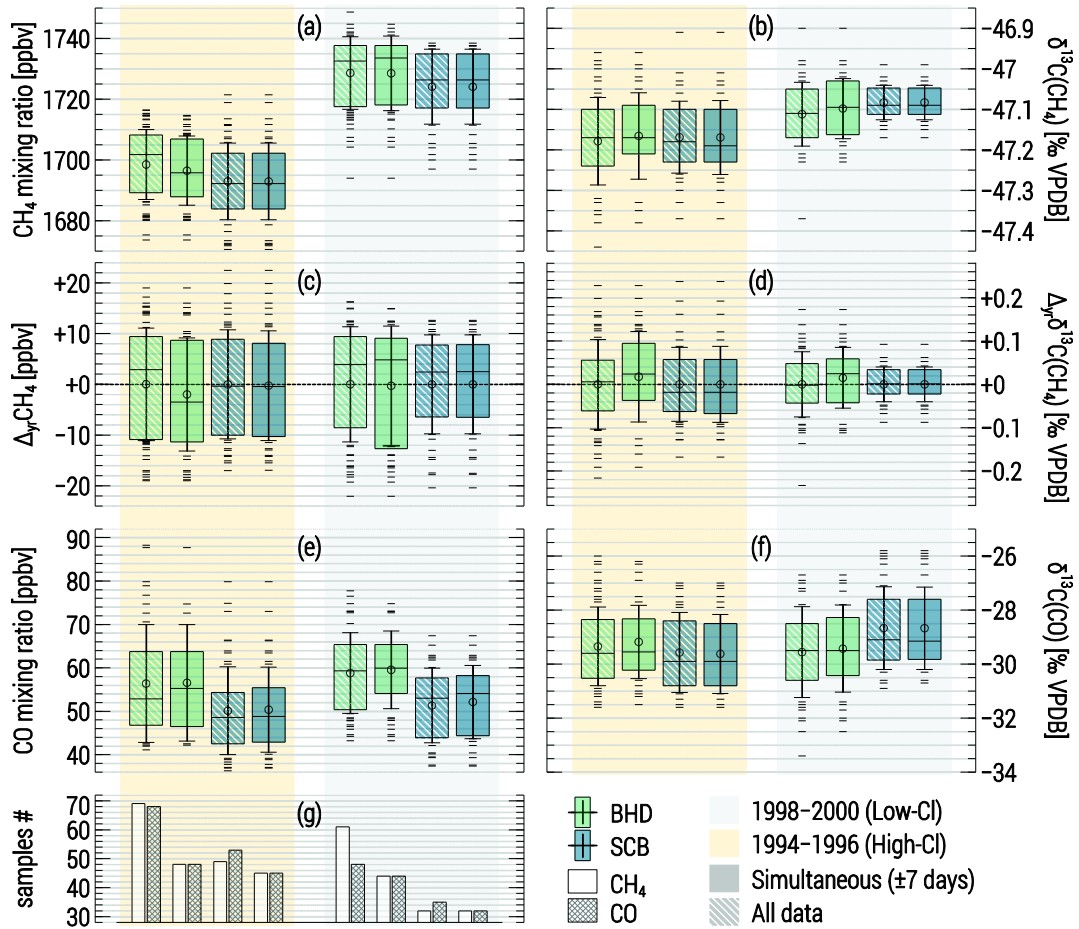

Fig. 1 Statistics on the CH₄ and CO mixing and $^{13}C/^{12}C$ ratios observed at Baring Head (BHD) and Scott Base (SCB) throughout the high-Cl (HC, orange shaded) and low-Cl (LC, grey shaded) periods hypothesised by Allan *et al.* (2007) (see text for details). Panels (**c, d**) show statistics on the anomalies with respect to the annual averages (denoted with "$\Delta_{yr}$"). Panel (**g**) displays the number of samples in each subset, respectively. The full time series of the data are shown in the Supplement (Fig. S2). Boxes and whiskers present the median/interquartile range and ±1σ (of the population) of the data. Circles and minus symbols denote the averages and samples falling outside ±1σ. Solid boxes denote the subset of data when CH₄ and CO samples were taken simultaneously (up to 7 days apart); hatched boxes refer to all data.

Table 1    Statistics on quasi-annual average (QAA) mixing/isotope ratios of CO observed/simulated at BHD and SCB.

| Data | Period | BHD | | | SCB | | |
|---|---|---|---|---|---|---|---|
| | | *n* | CO [nmol/mol] | $\delta^{13}C(CO)$ [‰] | *n* | CO [nmol/mol] | $\delta^{13}C(CO)$ [‰] |
| HC | 1994–1996 | 65 | 56.1 ±2.0 | −28.97 ±0.25 | 51 | 50.5 ±2.6 | −29.31 ±0.64 |
| LCª | 1998–2000 | 48 | 58.4 ±2.1 | −29.48 ±0.36 | 35 | 49.7 ±2.5 | −28.57 ±0.64 |
| Δ | HC−LC | | −2.2 ±2.9 | +0.51 ±0.43 | | +0.8 ±3.6 | −0.74 ±0.90 |
| | Significance (*p*-value)ᵇ | | | 0.12 / 0.002 | | | 0.79 / 0.28 |
| All data | 1989–2005 | 379(15/4) | 59.2 ±1.8 | −29.52 ±0.29 | 227(5/0) | 51.7 ±2.1 | −29.21 ±0.50 |
| | 1994–2005 | 192(5/1) | 57.8 ±2.1 | −29.38 ±0.36 | 155(0/0) | 50.8 ±2.3 | −29.13 ±0.58 |
| EMAC | 1996–2005ᶜ | | 57.0 ±3.5 | | | 51.3 ±1.7 | |
| | (incl. from CH₄ oxidation) | | 24.8 ±0.6 | | | 23.7 ±0.3 | |

Notes: Values in parentheses are the number of mild/extreme outliers (see the note4); the latter were excluded from the calculation of the long-term (up to 2005) averages. Quoted are standard errors of quasi-annual averages (±1σ).

a) Time-interpolated value is used for February (no samples are available at SCB during the LC period).

b) *p*-value is estimated for the null hypothesis that Δ of $\delta^{13}C(CO)$ QAA is below 0 / −2σ (left-tail test).

c) The aggregate of the emission inventories used in the simulation correspond closest to 2000 (see details in Gromov *et al.*, 2017).

## 2.3 EMAC model

[21] For extending the interpretation of observed ETSH CO, we resort to the results of simulations performed with the ECHAM5/MESSy Atmospheric Chemistry (EMAC) general circulation model (Jöckel *et al.*, 2010). EMAC includes all relevant processes (atmospheric transport, calculation of chemistry kinetics, photolysis rates, trace gas emissions, *etc.*) for simulating the current global atmospheric state. The setup we use resembles that of the EMAC evaluation study (MESSy Development Cycle 2, Jöckel *et al.*, 2010) and is augmented with kinetic tagging tools (Gromov *et al.*, 2010). These allow direct quantification of the CO component stemming from $CH_4$ oxidation (and as corollary provide $\lambda$) by following the carbon (C) exchanges through all intermediates (shown in Fig. S1) within a comprehensive chemistry mechanism simulated by the MECCA submodel (Module Efficiently Calculating the Chemistry of the Atmosphere, Sander *et al.*, 2011a). The emission setup contains only the standard emissions/precursors of Cl and yields average MBL Cl concentrations in the order of $10^1$–$10^2$ atoms cm$^{-3}$ (see the detailed simulated budgets in the Supplement, Table S1). These results are in line with MBL [Cl] of $(0.5$–$2) \times 10^2$ atoms cm$^{-3}$ obtained by Hossaini *et al.* (2016) in a similar model setup (ORG2).

[22] The QAAs of [CO] simulated in EMAC for the period 1996–2005 in the gridboxes enclosing the locations of BHD and SCB are also given in Table 1. Despite the spatial and temporal averaging used (~2.8° horizontal gridcell size at the T42L31-ECMWF resolution, weekly averages), model QAAs match observations well and have similar uncertainties (resulting from monthly means variation; the observed/simulated seasonalities are shown in the Supplement, Fig. S3). Due to longer lifetimes of CO and $CH_4$ in the well-mixed ETSH and, more importantly, their synchronous sink/production via OH, we expect much lower (factor $\sim 1/5$ compared to that of the total CO) variation in the $CH_4$-derived [CO] component. The fraction of the latter (denoted $\gamma$, see Table 2) is proportional to the average tropospheric $\lambda$ of 93% (diagnosed simulated value). Depending on the zonal domain, Cl atoms in EMAC initiate $(0.15$–$0.25)$% of $CH_4$ sink in the troposphere. The fraction of $CH_4$ removed in the ETSH (43 Tg(C) yr$^{-1}$) is minor compared to that in the tropics (271 Tg(C) yr$^{-1}$). About 13% of tropospheric sink occurs in the boundary layer.

[23] Additionally, we simulate the sink effective $^{13}$C enrichment in CO (denoted $\eta_c$) resulting from the $^{12}$C-preferential CO+OH reaction and removal of the $CH_4 \rightarrow CO$ chain intermediates (dry/wet deposition, when $\gamma < 1$), convoluted with atmospheric mixing and transport. The corresponding $\eta_c$ value at a given space-time point denotes how much higher the $\delta^{13}$C of airborne CO is compared to the case when sink KIEs were absent.[5] Altogether, values of $\gamma$ and $\eta_c$ at the stations and domain-wise integrals of $CH_4$ sink ($S$) and $\lambda$ (listed in Table 2) are used in the calculations that follow now.

## 2.4 Sensitivity of $\delta^{13}$C(CO) to the $CH_4$+Cl sink

[24] Using the observational and model data, we attempt to estimate the sensitivity of $\delta^{13}$C(CO) at a given station to supposed inter-annual changes in the Cl-initiated $CH_4$ sink. The QAA of $\delta^{13}$C(CO) (denoted $\delta_c$) can be approximated as a two-component mixture of $CH_4$- and non-$CH_4$-derived CO sources augmented by the effective sink enrichment:

$$\delta_c \cong (1 - \gamma)\delta_n + \gamma(\delta_m - \varepsilon_m) + \eta_c . \tag{1}$$

We refer the reader to Table 2 for the explanation of the parameters and their values. In essence, we account for the fractionations induced in atmospheric sinks ($\eta_c$ in CO and $\varepsilon_m$ in $CH_4$) and mix the sources in the proportion defined by $\gamma$. Exemplifying the estimate from A07, SH Cl changes should cause $\varepsilon_m$ to drop from 15‰ to 7‰ between the HC and LC, rendering $\delta^{13}$C of the carbon from $CH_4$ arriving to CO of $-62.2$‰ and $-54.2$‰, respectively. By rearranging Eq. (1) we derive the non-$CH_4$ CO source $\delta^{13}$C signature $\delta_n$ (see Table 2). Since there are virtually no surface sources of CO south of 40°S in the ETSH (see, *e.g.,* Gromov *et al.*, 2017, Sect. 3.4), the difference in $\delta_n$ at BHD and SCB could be driven only by poleward $^{13}$C-enrichment of the non-$CH_4$ in-situ sources (*e.g.* oxidation of higher hydrocarbons) and/or a stronger (than simulated in EMAC) zonal gradient in $\eta_c$. Note that the station-wise $\delta_n$ discrepancy scales with the $\varepsilon_m$ value, however not strongly: at $\varepsilon_m$ of OH sink KIE (3.9‰) it reduces from $(2.2\pm2.1)$‰ to $(1.5\pm2.2)$‰. In a statistical sense, the derived $\delta_n$ values reflect the same underlying source signature ($p$-value is 0.31).

---

[5] This value is obtained in a sensitivity simulation (*e.g.* without the KIEs in CO sink and removal of $CH_4 \rightarrow CO$ chain intermediates) and implies linearity (additivity) of atmospheric mixing and transport processes with respect to species $\delta^{13}$C (see details in Gromov (2013), Sects. 6.2.4–5).

**Table 2** Parameters used in calculus

| Species / Parameter [unit] | | Value | |
|---|---|---|---|
| CO | Station: | BHD | SCB |
| $\gamma^{\dagger}$ | CH$_4$-derived component [%] | 43±3 | 46±2 |
| $\eta_c^{\dagger}$ | Eff. $^{13}$C sink fractionation [‰] | +4.2±0.2 | +4.6±0.1 |
| $\delta_n^{*}$ | $\delta^{13}$C of non-CH$_4$ sources [‰] | −15.0±1.7 | −12.8±1.3 |
| $\delta_c$ | Observed $\delta^{13}$C(CO) [‰] | −29.5±0.3 | −29.2±0.5 |
| | | | |
| CH$_4$ | Domain: | SH | ETSH |
| $S^{\dagger,\S}$ | Total sink [Tg(C) yr$^{-1}$] | 187.8 | 52.5 |
| $\delta_m$ | Observed $\delta^{13}$C(CH$_4$) [‰] | | −47.2 |
| $\lambda^{\dagger}$ | Yield of CO from CH$_4$ | | 93% |
| | | | |
| | Period:$^{\ddagger}$ | HC | LC |
| $\Delta S$ | Changes to $S$ due to Cl variations [Tg(C) yr$^{-1}$] | +18 | 0 |
| $\varepsilon_m$ | Total CH$_4$ sink KIE [‰] | 15 | 7 |

Notes: Quoted QAAs and standard errors (±1σ); the latter are omitted for the components contributing to $\delta_c$ and $\delta_n$ errors insignificantly.
$^{\dagger)}$ Estimate based on EMAC results.
$^{*)}$ Derived at $\varepsilon_m = 11$‰ (average of the LC and HC periods).
$^{\S)}$ Includes the LC Cl sink term from A07 (9.7 Tg(C) yr$^{-1}$). For the SH, the sum of the ETSH and halved intra-tropical integrals is taken.
$^{\ddagger)}$ Estimates from A07.

[25] Using Eq. (1) defining $\delta_c$ in the HC and LC periods, one obtains its sensitivity ($\Delta\delta_c$) to changes in the CH$_4$+Cl sink ($\Delta S$) and in the total sink KIE ($\Delta\varepsilon_m$):[6]

$$\Delta\delta_c = (\lambda_a/\lambda)^{\mathrm{LC}}\gamma\left((\delta_m - {}^{\mathrm{HC}}\varepsilon_m - \delta_n)\mu - \Delta\varepsilon_m\right). \quad (2)$$

Here superscripts indicate the period the values are taken for, $\Delta$ denotes the HC−LC difference (same as in Sect. 2.2 above) and $\mu = \Delta S/^{\mathrm{LC}}S$ is the change in the total CH$_4$ sink $S$ relative to the LC conditions. The value of $S$ represents tropospheric column of a given domain, *i.e.* we assume that $\Delta S$ is distributed homogeneously over the SH or ETSH. Formulated using $\gamma$, Eq. (2) allows projecting the results for the alternative CO yield value $\lambda_a$ (different from that obtained in EMAC), as our simulations confirm that $\lambda$ directly proportionates $\gamma$ and $S$ in the tropospheric column (but not in the MBL). Furthermore, $\Delta\delta_c$ is derived under the assumption of constancy of $\eta_c$ and $\delta_n$ values. Whilst for $\eta_c$ such is likely the case (judging by the very similar observed CO mixing ratios, hence lifetimes, during HC and LC), for the latter an upper limit of ±1‰ can be put from the typical variation in the $\delta^{13}$C of the underlying sources (see Gromov *et al.*, 2017, Table 5). This is lower than the uncertainty associated with here derived $\delta_n$ values (*cf.* Table 2); we discuss the range of $\delta_n$ values required to concomitantly mask the changes in $\delta_c$ below.

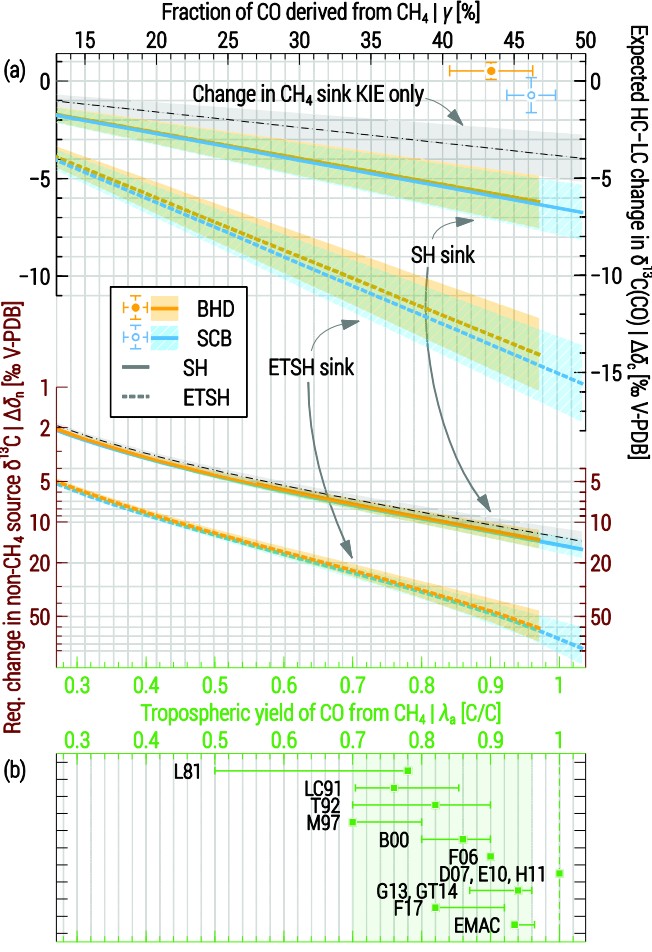

Fig. 2 (**a**) Top: Expected CH$_4$+Cl sink-driven changes to $\delta^{13}$C(CO) between HC−LC at the ETSH stations ($\Delta\delta_c$) as a function of CH$_4$-derived CO fraction ($\gamma$, top axis) resulting from assumed yield values ($\lambda_a$, bottom axis, approximate). Large symbols denote the observed (ordinate) and simulated (abscissa, EMAC) values. Thick lines present $\Delta\delta_c$ values calculated using Eq. (2) assuming that hypothesised changes to the CH$_4$+Cl sink occur within the entire SH (solid) and ETSH only (dashed). Thin dash-dotted lines exemplify the effect due to mere changes in CH$_4$ sink KIE ($\Delta\varepsilon_m$). Bottom: Average augmentation to the non-CH$_4$ sources signature ($\Delta\delta_n$) required to compensate $\Delta\delta_c$ at the respective values/domains (note the different axis shown in red). Errors bars/shaded areas denote ±1σ of the annual means/derived estimates. See Sects. 2.4 and 3 for details. (**b**) Tropospheric yield of CO from CH$_4$ oxidation reckoned in the current and previous studies. Symbols (error bars) denote the best (range of) estimates or the global (domain) averages. Abbreviations refer to: L81 – Logan *et al.* (1981), LC91 – Lieveld and Crutzen (1991), T92 – Tie *et al.* (1992), M97 – Manning *et al.* (1997), B00 – Bergamaschi *et al.* (2000), F06 – Folberth *et al.* (2006), D07 – Duncan *et al.* (2007), E10 – Emmons *et al.* (2010), H11 – Hooghiemstra *et al.* (2011), G13 – Gromov (2013), GT14 – Gromov and Taraborrelli, MPI-C (unpublished results using EMAC, 2014), F17 – Franco *et al.* (2017), EMAC – current study.

---

[6] Explicit derivation of this and following Eqs. is shown in Appendix A.

[26] Fig. 2 (**a**) shows the values of $\Delta\delta_c$, calculated for different stations/domains, as a function of $\gamma$ (implicitly scaling with arbitrarily chosen yield value $\lambda_a$). Very large changes are expected for the ETSH, where $\mu$ is about four times that in the SH. Importantly, the $^{LC}S$ value includes the Cl sink term from A07 (which is ~29 times greater than the total tropospheric $CH_4$+Cl sink simulated in EMAC), hence we receive the "lowest sensitivity" for the case when the Cl sink is added up to (instead of partly replacing) the other $CH_4$ sinks, *e.g.* that via OH. Alternatively, $\Delta\delta_c$ will additionally intensify by $-0.2$‰ and $-(1.8–2.1)$‰ in the SH and ETSH, respectively. By setting $\mu = 0$ in Eq. (2), we quantify the contribution of the $CH_4$ sink KIE (which increases by $\Delta\varepsilon_m$) only. Independent from the assumptions on the Cl sink domain and magnitude, it demonstrates the effect of lowering of $\delta^{13}C$ of C arriving to CO from $CH_4$ and accounts for $^1/_3$–$^2/_3$ of the total $\Delta\delta_c$ value (*cf.* Fig. 2, thin dashed line).

[27] Finally, we estimate the equivalent increase in the $\delta^{13}C$ value of the non-$CH_4$ sources ($\Delta\delta_n$) that would be required to mask the depleting effect of a hypothetical $CH_4$+Cl sink increase. We subtract Eq. (1) written for the HC and LC and solve it assuming $\Delta\delta_c = 0$ (notation from Eq. (2) is kept):

$$\Delta\delta_n = \frac{(\delta_n - (\delta_m - {}^{LC}\varepsilon_m))\mu + (1 + \mu)\Delta\varepsilon_m}{((\lambda_a/\lambda)^{LC}\gamma)^{-1} - (1 + \mu)}. \qquad (3)$$

Averages of $\Delta\delta_n$ at BHD/SCB are depicted in Fig. 2 (**a**) next to the black dots denoting the corresponding $\Delta\delta_c$ values. Similar to $\Delta\delta_c$, $\Delta\delta_n$ scales with the assumed domain and $CH_4$ input to CO, however stronger, because $\delta_n$ is closer to the $\delta^{13}C$ of the total CO source ($\delta_c - \eta_c$) as compared to that for $CH_4$ ($\delta_m - \varepsilon_m$). Thus, if we accept the EMAC-suggested tropospheric CO yield in the SH of $\lambda = 93\%$, Cl-driven changes to the $\delta^{13}C(CO)$ at BHD/SCB are expected to be of at least $-(5.8–6.3)$‰ between the LC and HC, unless these are masked by unrealistic concurrent increases in $\delta^{13}C$ of the non-$CH_4$ sources of about $+(11.6–13.5)$‰. If one assumes the $CH_4$+Cl sink changes only within the ETSH, these estimates scale to $-(13.1–14.5)$‰ and $+(46–61)$‰, respectively. It is important to note, that we gauge the expected changes to the annual averages of $\delta^{13}C(CO)$, which do integrate seasonal variations. The latter are observed at merely $\pm1.5$‰ (*cf.* Figs. 1 and S2) and should also increase strongly, if the Cl sink has a similar seasonal variation to that of OH (although A07 used a seasonal cycle based on DMS-related species in the SH, which has a shorter summer maximum).

## 3 Discussion

[28] The photochemical yield of CO from $CH_4$ constitutes a major factor of uncertainty in the CO budget. Modelling studies to date agreed on values of $\lambda \geq 0.7$ (see the overview in Fig. 2 (**b**)). Several recent studies (refs. D07, E10 and H11) suggest however $\lambda$ being close to unity and by doing so contradict findings of $^{13}CO$-inclusive studies (refs. M97, B00 and G13). Assuming that $\lambda < 0.7$ or that $\lambda \sim 1$ would be in conflict with basic principles, *i.e.* photochemical kinetics and/or dry and wet removal processes affecting the intermediates of the $CH_4 \rightarrow CO$ chain, or

their erroneous implementation in the global atmospheric models.

[29] Our estimates of $\Delta\delta_c$ bear the uncertainty of the assumed $\lambda$ value; nonetheless, they affirm that even if only 70% of reacted $CH_4$ molecules yield CO, at least one-third of the changes to the $\delta^{13}C$ signature of this source (that is, $(\delta_m + \varepsilon_m)$ times 0.7) should be expressed in the ETSH $\delta^{13}C(CO)$. Since $\delta_m$ changed by about $+0.1$‰ between the HC and LC periods (*cf.* Fig. 1 (**b**)), we conclude that $\varepsilon_m$ could not change by more than $+2$‰ in the SH as well (with this estimate being lower for $\lambda$ above 0.7).[7] Furthermore, statistically significant non-zero $\Delta\delta_c$ values (*p*-value of 0.01) should appear at very low $\lambda$, *viz.* above 0.05 (ETSH sink) and 0.12 (SH sink, respectively). We regard these two atmospheric domains because observations in the well-mixed ETSH may not single out the actual location of the Cl+$CH_4$ sink: The large part of sink-driven variations in mixing ratio and $\delta^{13}C$ of $CH_4$ and CO is merely transported into the ETSH from the tropics, where almost $^3/_4$ of the total $CH_4$ sink and accompanying CO production is expected (see Table S1 for EMAC results, also Gromov (2013), Sect. 6.2.3). Accordingly, Hossaini *et al.* (2016) also assign a major fraction of the $CH_4$+Cl sink to the lower latitudes. If such were not the case (*i.e.* varying Cl+$CH_4$ sink were confined to the ETSH), the estimated effect on $\delta^{13}C(CO)$ would roughly be twice that reckoned for the SH, *i.e.* extreme values.

[30] There are a few remarks on the usability of the method used by A07, in addition to the thorough theoretical enquiry by Lassey *et al.* (2011). Evidence, or at least indications, for Cl in the ETSH is based on the [$CH_4$] *vs.* $\delta^{13}C(CH_4)$ Lissajous (a.k.a. phase-) diagrams being ellipses in the case of seasonal cycles. The slope of their major axis gives the "apparent" KIE, from which the ratio Cl/OH can be inferred knowing the individual KIEs. Clearly, Cl was not assessed on the basis of the annual average value of $\delta^{13}C(CH_4)$ but on the basis of its seasonal cycle, which is small. Using annual averages, however, is yet impeded by perceptible long-term trends in [$CH_4$] and $\delta^{13}C(CH_4)$, which neither A07 (who consider the final 8 equilibrated years of the 40-year spin-up simulations) nor Lassey *et al.* (2011) (who use a rather idealised model) have accounted for. For example, presence and asynchronous evolution of [$CH_4$] and $\delta^{13}C(CH_4)$ long-term trends could result in different mixing and transport of $CH_4$ isotopologues compared to that resulting from trend-free simulated seasonal variations. We note that whereas observed [$CH_4$] growth is similar throughout both HC and LC periods, such is not the case for $\delta^{13}C(CH_4)$ which does not increase in the LC (*cf.* Fig. S2 (**a**, **c**) and, in particular, the seasonal time series fits for $CH_4$ at the NIWA website[8]). Furthermore, the latter is likely a global signal of the 2000–2007 intermittent stop in tropospheric $CH_4$ growth, which manifested itself in $\delta^{13}C$ earlier than in mixing ratios and terminated with the reversed $^{13}C/^{12}C$ trend (see, *e.g.*, Nisbet *et al.*, 2016). Currently available observational data do not allow unambiguous attribution of this global phenomenon to one or several causes proposed (Turner *et al.*, 2017), however.

[31] Our incomplete information about the $^{13}C$ isotopic composition of $CH_4$ sources presently prevents to single out a Cl-induced input into the annual average value of $\delta^{13}C(CH_4)$, even though it

---

[7] Calculated as $(\Delta\delta^{13}C(CO)-0.1$‰$)/(\gamma\cdot\lambda)$ for values at SCB (see Tables 1 and 2).

[8] https://www.niwa.co.nz/atmosphere/our-data/trace-gas-plots/methane (last access: December 2017).

should be perceptible (about +1.5‰, assuming for the sake of matter a 2.5% Cl sink). The corresponding negative shift in $\delta^{13}C(CO)$ is about 1.6‰ (estimated in Sect. 2.1). In this respect, $\delta^{13}C(CH_4)$ and $\delta^{13}C(CO)$ are equally sensitive to Cl. Because oxidation of $CH_4$ is a main source of CO in the ETSH, and the isotopic composition of atmospheric $CH_4$ is better known than that of its sources, it may well be that variation in the annual average value of $\delta^{13}C(CO)$ is more useful variable for estimating [Cl]. The relatively long lifetime and small seasonality in sources result in weak seasonal cycles of mixing ratio and $\delta^{13}C$ in $CH_4$. In contrast, the seasonal cycle of $\delta^{13}C(CO)$ is dominated by the large difference in isotopic composition of its sources, with the main driver being the switch between CO from $CH_4$ oxidation and that of the other sources. Since the presence of Cl makes $CH_4$ oxidation an even more $^{13}C$-depleted source, the impact of $CH_4$ oxidation on CO in the ETSH peaks and may render the seasonal amplitude/summer minima of $\delta^{13}C(CO)$ a sensitive indicator for Cl. Unfortunately, deficit of observational data (large uncertainties due to insufficient statistics) currently hinder such application.

[32] A fundamental problem remains that the ETSH $\delta^{13}C(CO)$ budget cannot be closed even when a Cl sink is excluded, unless a CO yield from $CH_4$ of 0.7–0.86 is assumed (Manning *et al.*, 1997, Bergamaschi *et al.*, 2000). Yields below unity leave however the possibility that a positive fractionation in the removal of the $CH_4 \rightarrow CO$ intermediates may be at play. Using $\lambda = (0.7$–$0.86)$ and $\gamma = 0.3$ for the troposphere, one calculates that an average KIE of (11–33)‰ should escort the removal of intermediates in order to offset the Cl input to $\delta^{13}C(CO)$. This estimate is 3–8 times higher than current parameterisations suggest (about 4‰, see Gromov, 2013, Sect. 6.2.4) and is even higher in the SH, where $\gamma$ is above 0.4. Another complication is potentially present because one cannot exclude, that the room temperature laboratory data for the $^{13}C$ KIE for CO+OH reaction are not applicable to the bulk of the troposphere, even though the reaction itself is little temperature- but mostly pressure-dependent (see Gromov, 2013, Sect. 6.1.4). The unbalanced $^{13}C(CO)$ budget may then be the consequence of underestimating the CO sink KIE in the models, despite adequate estimates of the sources' $^{13}C/^{12}C$ ratios.

## 4 Conclusions

[33] We emphasise the value of long-term observations of CO isotopic composition, especially at locations like Scott Base (Antarctica), where influence of local sources is least and the fraction of photochemically produced CO is largest. In combination with modelling (*e.g.* EMAC), $\delta^{13}C(CO)$ allows monitoring for intra-annual changes in the carbon isotopic composition of $CH_4$-derived CO, namely the $\delta^{13}C$ value of reacted $CH_4$ modified by the total sink KIE ($\varepsilon_m$). Within the range of probable $\lambda$ values (0.7−0.93), we are able to cap the potential changes in $\varepsilon_m$ by +(2.0–1.5)‰ between 1994−1996 and 1998−2000 in the ETSH, which contrasts the +8‰ derived by Allan *et al.* (2007). Conversely, $\delta^{13}C(CO)$ may also be employed for "top-down" estimates of $\delta^{13}C$ values of $CH_4$ sources, provided the $\varepsilon_m$ is equilibrated on a scale of tropospheric $CH_4$ lifetime. This could be achieved in a differential mixing model (also known as the

"Keeling" plot) contrasting little varying $CH_4$-derived [CO] and $\delta^{13}C$ and largely varying input from other CO sources (*e.g.* biomass burning).

[34] We conclude that $\delta^{13}C(CO)$ is particularly sensitive to the $CH_4$+Cl sink. Its temporal variations, if they exist, may allow to calibrate an independent "bottom-up" [Cl] proxy, *e.g.* emissions of Cl simulated in process-based models. For example, changes in observed $\delta^{13}C(CO)$ at SCB (see Table 1) allow variations of the Cl-driven sink of $CH_4$ not larger than $(1.5\ \lambda_a^{-1})$% of its total (assuming the yield $\lambda_a$ of CO from $CH_4$). Projecting this figure onto EMAC results (Table S1, zonal tropospheric integrals) implies that variations in mean ETSH chlorine abundance should have not exceeded $\Delta[Cl] = (0.9\ \lambda_a^{-1}) \times 10^3$ atoms $cm^{-3}$ between 1994–1996 and 1998–2000. Regarding the fact that Manning *et al.* (1997) and Bergamaschi *et al.* (2000) could only close the SH $^{13}C(CO)$ budget assuming $\lambda$ values of 0.7 and 0.86, which are within the generally accepted range, it is unlikely that tropospheric Cl is as high as assumed in the literature.

[35] Although invoking isotopic information often is like opening a can of worms (scientists' favourite diet), relevant conclusions emerge. Lassey *et al.* (2011) exposed shortcomings of the phase diagram method; we show here, using a low- and high-Cl scenario, that unrealistic yield values of CO from $CH_4$ oxidation ($\lambda$ below 0.12 in the SH) and/or implausible increases in the $\delta^{13}C$ of non-$CH_4$ sources of CO (exceeding +7‰ at realistic $\lambda \geq 0.7$) would have to be assumed to explain the absence of concurrent inter-annual variations in $\delta^{13}C(CO)$ in the ETSH. This constitutes an independent, observation-based evaluation of [Cl] variations envisaged by Allan *et al.* (2007), from which we conclude that such variations are extremely unlikely. Concerning estimates of background levels of Cl, even attributing 1% of the total tropospheric sink of $CH_4$ to Cl aggravates the non-trivial problem of balancing the global $^{13}C(CO)$ budget. It follows that the role of tropospheric Cl as a sink of $CH_4$ oxidation (see, *e.g.*, Saunois *et al.*, 2016, and refs. therein) is seriously overestimated.

## Code availability

[36] The Modular Earth Submodel System (MESSy) is continuously further developed and applied by a consortium of institutions. The usage of MESSy (including the EMAC model) and access to the source code is licenced to all affiliates of institutions which are members of the MESSy Consortium. Institutions can become a member of the MESSy Consortium by signing the MESSy Memorandum of Understanding. More information can be found on the MESSy Consortium Website (http://www.messy-interface.org).

## Appendix A. Derivations

[37] Below we detail the derivation of Eqs. (2) and (3). The former is obtained by writing Eq. (1) for the HC and LC periods:

$$^{HC}\delta_c \cong (1 - {}^{HC}\gamma)\delta_n + {}^{HC}\gamma(\delta_m - {}^{HC}\varepsilon_m) + \eta_c\ ,$$

$$^{LC}\delta_c \cong (1 - {}^{LC}\gamma)\delta_n + {}^{LC}\gamma(\delta_m - {}^{LC}\varepsilon_m) + \eta_c\ ,$$

and subtracting these to yield the respective change to $\delta_c$:

$$^{HC}\delta_c - {}^{LC}\delta_c = ({}^{HC}\gamma - {}^{LC}\gamma)(\delta_m - \delta_n) - {}^{HC}\gamma{}^{HC}\varepsilon_m + {}^{LC}\gamma{}^{LC}\varepsilon_m\ .$$

Note that $\gamma$ is proportional to the product ($\lambda \cdot S$) and hence increases by $(1 + \Delta S/^{LC}S)$ during the HC period. Thus, using

$$\mu = \Delta S/^{LC}S \,,$$
$$^{HC}\gamma/^{LC}\gamma = (1 + \mu) \,,$$
$$\Delta\varepsilon_m = {}^{HC}\varepsilon_m - {}^{LC}\varepsilon_m \,,$$

and factoring with respect to $^{LC}\gamma$, one obtains:

$$\Delta\delta_c = {}^{HC}\delta_c - {}^{LC}\delta_c = {}^{LC}\gamma((\delta_m - {}^{HC}\varepsilon_m - \delta_n)\mu - \Delta\varepsilon_m) \,.$$

610 Finally, the value of $\Delta\delta_c$ can be projected for any arbitrary yield value $\lambda_a$ (different to $\lambda$ obtained in EMAC and used in our calculations) by scaling the value of $^{LC}\gamma$ with ($\lambda_a/\lambda$), which yields Eq. (2).

[38] Derivation of Eq. (3) is done in a similar fashion, *i.e.* equat-
615 ing the right hand sides of Eq. (1) written for HC and LC periods (assuming that $\delta_c$ does not change):

$$(1 - {}^{LC}\gamma)\delta_n + {}^{LC}\gamma(\delta_m - {}^{LC}\varepsilon_m) =$$
$$(1 - {}^{LC}\gamma(1 + \mu))(\delta_n + \Delta\delta_n) + {}^{LC}\gamma(1 + \mu)(\delta_m - ({}^{LC}\varepsilon_m + \Delta\varepsilon_m)) \,.$$

Rearranging the above expression for $\Delta\delta_n$ (required change in $\delta_n$ sought) and factoring with respect to $^{LC}\gamma$ yields:

$$\Delta\delta_n = \frac{(\delta_n - (\delta_m - {}^{LC}\varepsilon_m))\mu + (1 + \mu)\Delta\varepsilon_m}{({}^{LC}\gamma)^{-1} - (1 + \mu)}$$

where $^{LC}\gamma$ can be further modulated by ($\lambda_a/\lambda$) to account for an
620 arbitrary yield value, as shown in Eq. (3).

## Acknowledgements

[39] We are grateful to Martin Manning, Taku Umezawa and Sander Houweling for discussions on the isotopic composition of CH$_4$. The unique long-term trace gas records from Baring Head (New Zealand) and Scott Base (Antarctica) made available
625 by National Institute of Water and Atmospheric Research (NIWA, 2010) are of great value. We also thank Roland Eichinger for useful comments during the manuscript preparation.

[40] This work was supported by German Federal Ministry of Ed-
630 ucation and Research (BMBF) as Research for Sustainability initiative (FONA, http://www.fona.de) through PalMod project (FKZ: 01LP1507A).

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

**Supplement**

# A very limited role of tropospheric chlorine as a sink of the greenhouse gas methane

Sergey Gromov *et al.*

Correspondence to: Sergey Gromov (sergey.gromov@mpic.de)

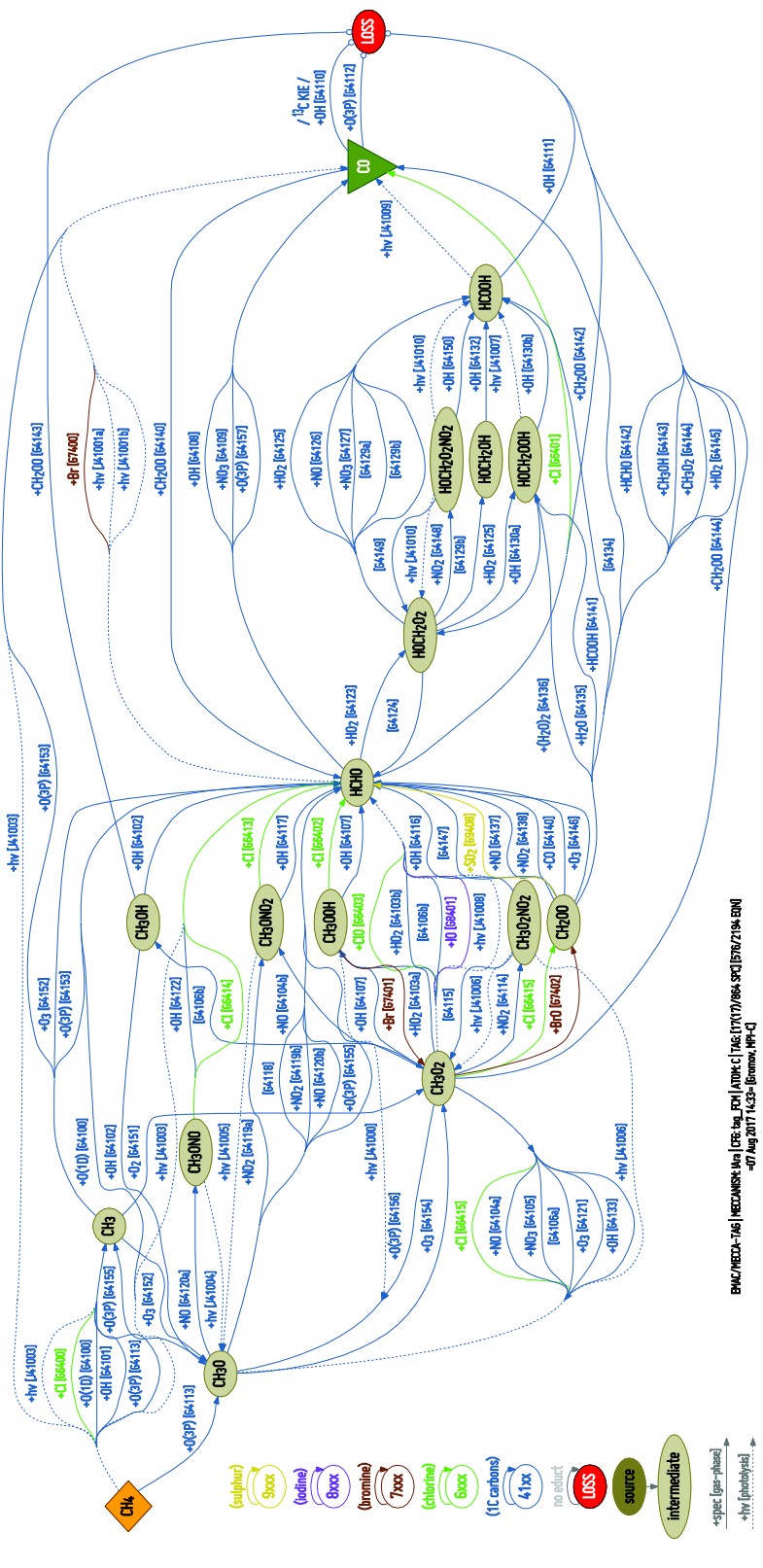

Fig. S1 Diagram of reaction pathways (following C exchanges) between CH$_4$ and CO as simulated in MECCA (the kinetic chemistry submodel used in EMAC, see Sect. 2.3 of the manuscript for details). Each arrow denotes a single gas-phase (solid line) or photolysis (dashed line) reaction; caption lists the reaction partner and label; colours refer to the chemical mechanism groups defined in MECCA. Pathways ending at the loss reservoir remove C from the CH4 → CO conversion chain. Note that non-chemical removal of C from the system (e.g. dry/wet deposition of CH$_3$O$_2$, CH$_3$OH, HCHO, HCOOH intermediates) is not shown, however, simulated by the model. See Supplement to Lelieveld et al. (2016) (p. S18, https://www.atmos-chem-phys.net/16/12477/2016/acp-16-12477-2016-supplement.pdf) for the complete listing of the respective MECCA reaction mechanism.

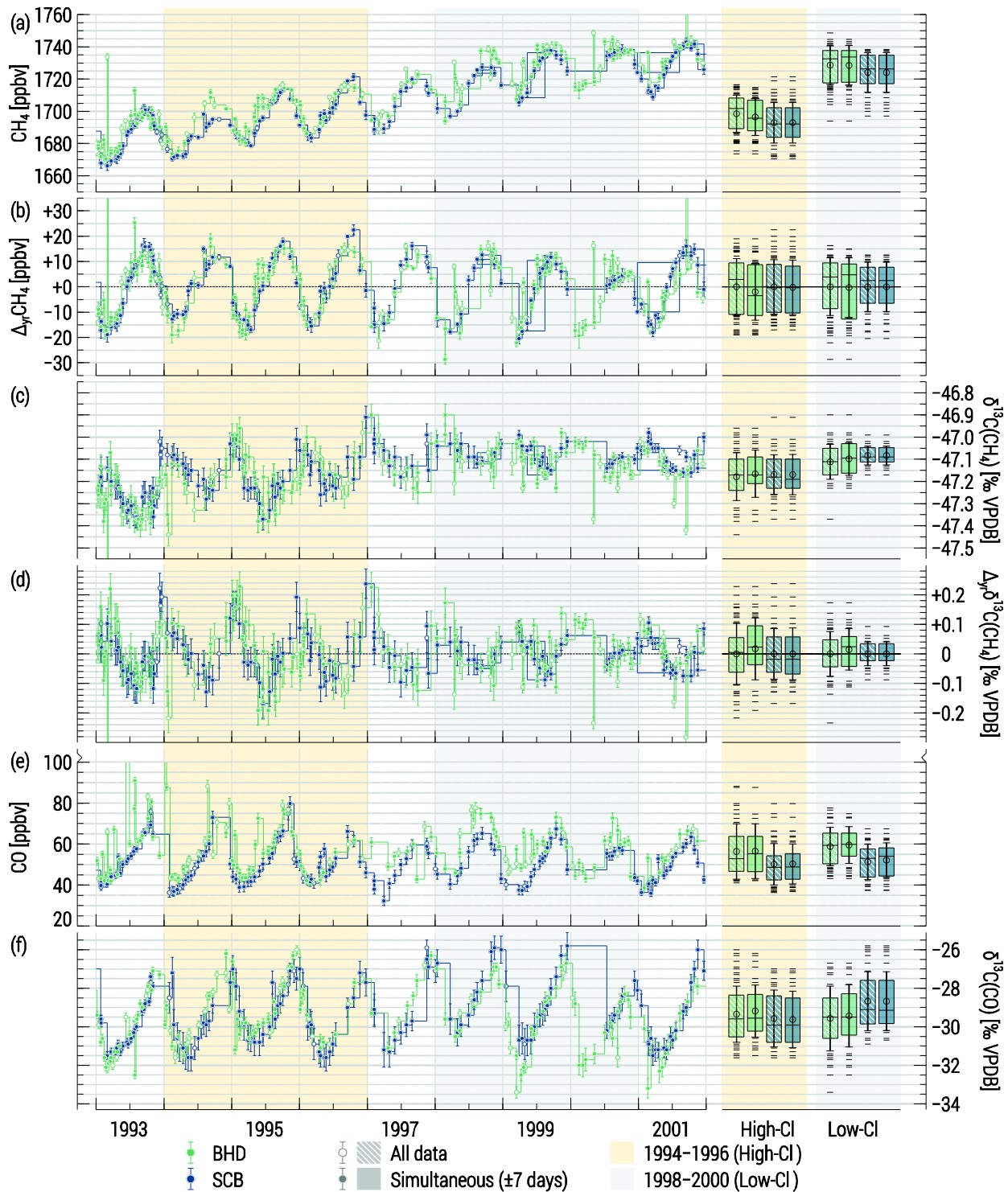

Fig. S2 Time series (left) and statistics (right, box-and-whisker plots) of the observations from Baring Head (BHD) and Scott Base (SCB) scrutinised in this study. Panels (**a, c**) present the mixing ratios and $\delta^{13}C$ of $CH_4$; panels (**b, d**) show anomalies with respect to the annual averages (denoted with "$\Delta_{yr}$"). Panels (**e, f**) display the mixing ratios and $\delta^{13}C$ of CO. The number of samples in each subset is presented in the manuscript (Fig. 1, panel (**g**)). Shaded areas denote the ETSH MBL high-Cl (orange shaded) and low-Cl (grey shaded) periods hypothesised by A07 (see text for details). Step lines navigate through the entire time series at each station. Boxes and whiskers present the median/interquartile range and $\pm 1\sigma$ (of the population) of the selected data. Circles and minus symbols denote the averages and samples falling outside $\pm 1\sigma$, respectively. Solid symbols/boxes refer to the data when $CH_4$ and CO samples were taken simultaneously (up to 7 days apart); hollow symbols/hatched boxes refer to all data.

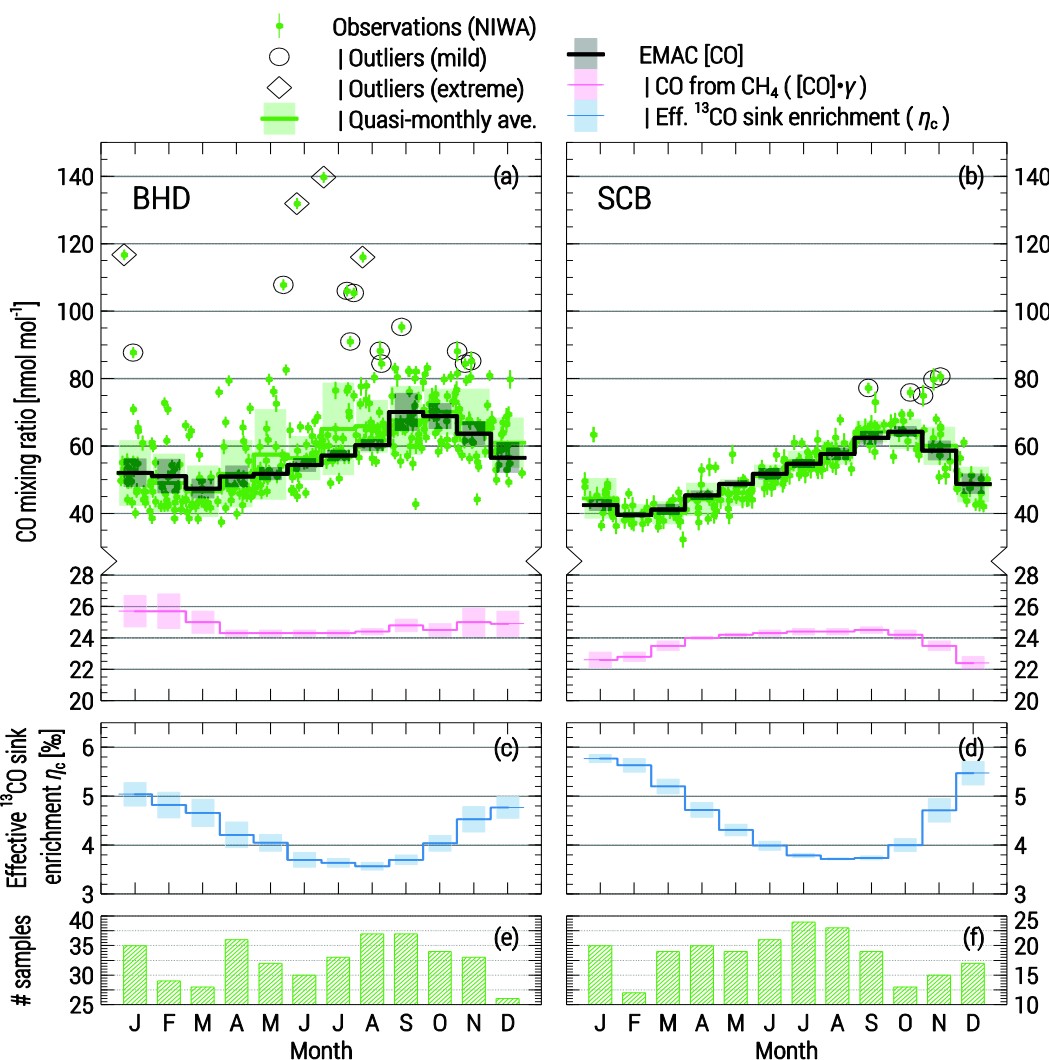

Fig. S3 Seasonal cycles CO mixing ratio at Baring Head (BHD, panel **a**) and Scott Base (SCB, panel **b**). Observations (entire data series plotted against day of year) are shown with symbols; circles and diamonds denote mild and extreme outliers (see Sect. 2.2 of the manuscript for details). Step lines refer to quasi-monthly averages derived from the observations (green) and from the EMAC model (1996–2005) for total CO (black) and its component derived from $CH_4$ oxidation (thin red line, lower scale). Panels (**c, d**) present the simulated effective sink $^{13}$CO enrichment, respectively. Vertical bars indicate $\pm 1\sigma$ of the subsample used for quasi-monthly averages. Panels (**e, f**) show the number of samples in observational data. Mind the breaks and different scales of the ordinate axes.

Table S1 Annual average CO- and CH4-related integrals by domain simulated in EMAC for 1996–2005.

**CO burden [Tg(C)]**

| vertical | GLOB | ETNH | IT | ETSH | AR | AN |
|---|---|---|---|---|---|---|
| SRF | 1.80 | 0.70 | 0.86 | 0.24 | 0.09 | 0.03 |
| BL | 12.71 | 5.07 | 5.87 | 1.77 | 0.52 | 0.19 |
| MBL | 6.39 | 1.67 | 3.31 | 1.40 | 0.20 | 0.05 |
| FT | 135.71 | 46.82 | 68.13 | 20.75 | 7.45 | 2.42 |
| T | 151.30 | 53.09 | 75.23 | 22.98 | 8.13 | 2.67 |
| TP | 4.89 | 2.14 | 1.67 | 1.08 | 0.48 | 0.19 |
| LMS | 15.64 | 6.71 | 4.25 | 4.68 | 1.45 | 0.84 |

**Fraction of CO from CH4 oxidation | $\gamma$ [C/C]**

| vertical | GLOB | ETNH | IT | ETSH | AR | AN |
|---|---|---|---|---|---|---|
| SRF | 24.6% | 17.1% | 25.2% | 44.8% | 19.3% | 46.0% |
| BL | 25.7% | 17.5% | 27.2% | 44.5% | 18.9% | 45.8% |
| MBL | 32.5% | 19.9% | 33.5% | 45.2% | 19.0% | 45.8% |
| FT | 29.5% | 21.2% | 31.0% | 43.6% | 20.1% | 45.4% |
| T | 29.2% | 20.7% | 30.6% | 43.7% | 20.0% | 45.5% |
| TP | 30.5% | 23.9% | 32.8% | 40.3% | 21.8% | 43.6% |
| LMS | 40.7% | 33.3% | 44.7% | 47.5% | 30.5% | 49.5% |

**Cl concentration [atoms cm$^{-3}$]**

| vertical | GLOB | ETNH | IT | ETSH | AR | AN |
|---|---|---|---|---|---|---|
| SRF | 97 | 56 | 145 | 47 | 8 | 10 |
| BL | 100 | 59 | 152 | 51 | 8 | 11 |
| MBL | 110 | 59 | 165 | 49 | 8 | 9 |
| FT | 269 | 163 | 352 | 152 | 36 | 42 |
| T | 261 | 157 | 345 | 146 | 35 | 40 |
| TP | 3151 | 1415 | 4945 | 1418 | 299 | 506 |
| LMS | 16172 | 15892 | 15630 | 17260 | 15301 | 20021 |

**Yield of CO from CH4 oxidation | $\lambda$ [C/C]**

| vertical | GLOB | ETNH | IT | ETSH | AR | AN |
|---|---|---|---|---|---|---|
| SRF | 72% | 72% | 74% | 62% | 48% | 51% |
| BL | 81% | 83% | 81% | 73% | 55% | 62% |
| MBL | 47% | 41% | 45% | 63% | 103% | 257% |
| FT | 96% | 97% | 95% | 96% | 98% | 107% |
| T | 93% | 95% | 93% | 93% | 96% | 104% |
| TP | 123% | 119% | 128% | 121% | 120% | 120% |
| LMS | 102% | 103% | 101% | 103% | 103% | 102% |

**CH4 sink | S [Tg(C) yr$^{-1}$]**

| vertical | GLOB | ETNH | IT | ETSH | AR | AN |
|---|---|---|---|---|---|---|
| SRF | 7.0 | 1.1 | 5.2 | 0.69 | 0.01 | 0.01 |
| BL | 49.2 | 7.9 | 36.1 | 5.1 | 0.07 | 0.06 |
| MBL | 32.7 | 2.9 | 25.9 | 3.9 | 0.03 | 0.01 |
| FT | 313.8 | 49.5 | 227.8 | 36.5 | 2.22 | 1.01 |
| T | 372.7 | 59.2 | 270.6 | 42.8 | 2.32 | 1.09 |
| TP | 1.83 | 0.59 | 0.74 | 0.50 | 0.07 | 0.05 |
| LMS | 20.2 | 5.5 | 9.1 | 5.59 | 0.67 | 0.67 |

**(via OH) [C/C]**

| vertical | GLOB | ETNH | IT | ETSH | AR | AN |
|---|---|---|---|---|---|---|
| SRF | 99.9% | 99.9% | 99.9% | 99.8% | 99.8% | 99.8% |
| BL | 99.9% | 99.9% | 99.9% | 99.8% | 99.8% | 99.8% |
| MBL | 99.9% | 99.9% | 99.9% | 99.8% | 99.8% | 99.8% |
| FT | 99.7% | 99.7% | 99.7% | 99.7% | 99.7% | 99.6% |
| T | 99.7% | 99.7% | 99.7% | 99.7% | 99.7% | 99.6% |
| TP | 90.9% | 94.2% | 86.4% | 93.7% | 97.8% | 96.5% |
| LMS | 53.4% | 58.7% | 47.9% | 57.1% | 64.7% | 60.4% |

**(via O$^1$D) [C/C]**

| vertical | GLOB | ETNH | IT | ETSH | AR | AN |
|---|---|---|---|---|---|---|
| SRF | 0.01% | 0.01% | 0.01% | 0.01% | 0.02% | 0.04% |
| BL | 0.01% | 0.02% | 0.01% | 0.01% | 0.02% | 0.05% |
| MBL | 0.01% | 0.01% | 0.01% | 0.01% | 0.02% | 0.03% |
| FT | 0.07% | 0.08% | 0.07% | 0.07% | 0.06% | 0.09% |
| T | 0.06% | 0.07% | 0.06% | 0.06% | 0.06% | 0.09% |
| TP | 1.94% | 1.13% | 3.20% | 1.05% | 0.38% | 0.40% |
| LMS | 33.0% | 26.8% | 40.3% | 27.1% | 18.6% | 18.4% |

**(via Cl) [C/C]**

| vertical | GLOB | ETNH | IT | ETSH | AR | AN |
|---|---|---|---|---|---|---|
| SRF | 0.13% | 0.11% | 0.13% | 0.14% | 0.14% | 0.19% |
| BL | 0.13% | 0.12% | 0.13% | 0.15% | 0.15% | 0.19% |
| MBL | 0.14% | 0.13% | 0.14% | 0.15% | 0.15% | 0.18% |
| FT | 0.25% | 0.24% | 0.25% | 0.27% | 0.22% | 0.30% |
| T | 0.23% | 0.22% | 0.23% | 0.25% | 0.22% | 0.29% |
| TP | 7.2% | 4.7% | 10.4% | 5.2% | 1.8% | 3.1% |
| LMS | 13.7% | 14.6% | 11.8% | 15.8% | 16.7% | 21.2% |

**Domain abbreviations**

| | | | | |
|---|---|---|---|---|
| zonal: | GLOB | Globe (90°S–90°N) | NH/SH | Northern/Southern Hemisphere |
| | IT/ET | Intra/Extra-Tropics (separated at 23.4°N/°S) | AR/AN | Arctic/Antarctic (above 66°N/°S) |
| vertical: | SRF/TP | Surface (lowest model layer) | T | Troposphere (below the TP) |
| | (M)BL | (Marine) Boundary Layer | FT | Free Troposphere (above the BL, below the TP) |
| | TP | Tropopause | LMS | Lowermost Stratosphere (above the TP) |