# Peer review of "A very limited role of tropospheric chlorine as a sink of the greenhouse gas methane"

_Atmospheric Chemistry and Physics, 2018_

## Referee Comment (RC1) · M. Manning (Referee) · 23 Mar 2018

Comments on "A very limited role of tropospheric chlorine as a sink of the greenhouse gas methane" by S. Gromov *et al*

Martin Manning (Referee)

martin.manning@vuw.ac.nz

**General Comments**

Given the number of recent papers that have proposed quite different explanations for the post-2006 rise in atmospheric $CH_4$, it is clear that there are still some major systemic uncertainties in our understanding of the $CH_4$ source – sink budget. While the increasing amount of isotopic ($\delta^{13}CH_4$) data should help to resolve these uncertainties, this has to deal with a limited understanding of removal by Cl with its very large kinetic isotope effect (KIE), and so a significant effect on $\delta^{13}CH_4$ even though most of $CH_4$ removal is by OH.

Analyses by Allan *et al* (2005, 2007) [NB additional references not included in the Gromov *et al* paper are given below] suggested that removal by Cl in the marine boundary layer can match $\delta^{13}CH_4$ data in the Southern Hemisphere, so long as there is a significant amount of interannual variability in the amount of this removal, but the driving factors for such variability in Cl still needed to be clarified. The Lassey *et al* 2011 analysis was then based on a simpler budget approach but showed that small interannual variations in the seasonal cycles for different sources can also lead to an 'apparent KIE' in the data that is quite different to that due to chemical removal processes alone.

More recent work by Hossaini *et al* (2016) has shown that, when a detailed form of tropospheric Cl chemistry is added to the TOMCAT chemical transport model, that sink for $CH_4$ appears to be about half what was used in Allan *et al*, but that there is also the potential for large scale regional effects and higher amounts of Cl in some places.

This new paper by Gromov *et al* is definitely a very important extension to the work cited above, because it now addresses the issue of how a highly fractionating removal of $CH_4$ would affect the atmospheric CO that is produced by both of the $CH_4$ + OH and $CH_4$ + Cl removal processes. A key point made in this paper is that the role of Cl in removal of $CH_4$ must be kept consistent with data and budget analyses for $\delta^{13}CO$, and that the long records of NIWA data in the Southern Hemisphere are very relevant for this. In addition, because $CH_4$ oxidation produces 40 - 50% of the CO that is observed in the Southern Hemisphere, the isotopic effects of removal by Cl should be more evident there.

The EMAC model that is used in this analysis, and the atmospheric chemistry that it covers, are quite well documented in a number of earlier papers and the tagging tools, described in Gromov *et al* 2010, provide a clear way of attributing CO to its different sources. So, the framework used in this paper has a clear basis.

However, there are some aspects of the paper that I find to be either not clear or incomplete as follows.

1) Use of the EMAC model in this work appears to have constant surface sources for CO over the 1994 – 2000 period and so does not include the effects of any trends or interannual source variations. Because the lifetime of CO is about forty times shorter than that for $CH_4$, its mixing ratio and $\delta^{13}CO$ are much more sensitive to interannual variations in its sources. In particular, biomass burning is a significant source of CO in the Southern Hemisphere and its interannual variations are not well known prior to 1996 (e.g. Giglio *et al* 2013). More

generally, while burning in C4 ecosystems is known to be dominant, the interannual variations are larger for C3 ecosystems that have a quite different $\delta^{13}C$ (e.g. Randerson *et al* 2005) so there can be relatively larger interannual variability in the source's $\delta^{13}C$ than in its magnitude.

Southern Hemisphere data at the start of the 1998 – 2000 period will also be affected by the extensive biomass burning emissions from Indonesia that continued for a longer period than usual in 1997. This is seen in the NIWA data that have an (admittedly) noisy long-term maximum in CO mixing ratio in 1998 but a much clearer maximum in $\delta^{13}CO$ that year. It is still not clear to what extent these source variations will affect the relationships in the tightly coupled $CH_4$ – $CO$ – $OH$ system, but as noted in Lassey *et al*, 2011, the 'apparent KIE' for $CH_4$ is quite sensitive to variations in seasonal cycles for the sources.

Some structural differences between the two 3-year periods used in this paper are seen quite clearly in Fig S2, e.g. the much smaller amplitude for $\delta^{13}CH_4$ seasonal cycles over 1998 – 2000 as shown in Fig S2(d). Therefore, it would be better to show all of the data this way in the main text, rather than just the statistical summaries currently used in Fig 1. Similarly, Fig S3 is a very clear way of showing model results and would also be useful in the main text.

To summarise this point: given that CO and $\delta^{13}CO$ are much more sensitive to changes in sources than $CH_4$ and $\delta^{13}CH_4$, it is important to consider how the paper's use of a fixed non-$CH_4$ source for CO may have hidden some of the differences between the two periods.

2) It appears from this paper, and from the earlier Gromov (2013), that the EMAC model has produced CO mixing ratio values but not the $\delta^{13}CO$ values directly and that is why there is no analogue for Fig S3 showing model results for $\delta^{13}CO$ as well. Instead, section 2.4 gives formulae that bring together model and observational data, as summarised in Table 2, and then introduce the comparison between high Cl and low Cl periods shown in Fig 2.

However, while equation (1) is valid when $CH_4$ and CO are at equilibrium with their average sources and sinks, it does not apply more generally for the seasonal cycles in $CH_4$ and CO mixing ratios and $\delta^{13}Cs$. Neither the mixing ratios nor $\delta^{13}Cs$ are at equilibrium due to the significant seasonal cycles in sources and removal rates. And, while large differences in lifetimes mean that the dis-equilibrium will be larger for $CH_4$ than for CO, in both cases the isotope ratios are expected be slower at reaching equilibrium than mixing ratios (e.g. Tans, 1997). Furthermore, the tightly coupled $CH_4$ – $CO$ – $OH$ system has different modes of variation (Prather, 1994, etc) and these are different again for the isotope ratios (Manning, 1999).

This difference between an equilibrium and dynamic situation appears when comparing the change in $\delta^{13}CH_4$ corresponding to equilibrium conditions for the HC and LC periods, as derived in section 2.4, with results shown in Fig 2b of Allan *et al* (2007), based on runs with the UK Met Office Unified Model, for the same estimates of Cl concentrations. The Allan et al difference in $\delta^{13}CH_4$ between the two different Cl concentrations is less than half that given in section 2.4. This point is mentioned again in specific comments on lines 181 – 193 below.

Therefore, it is not clear to what extent seasonal cycles in the dis-equilibrium and differences in that for both $CH_4$ and CO as well as for mixing ratios and $\delta^{13}Cs$, will modify

what is summarised in section 2.4. A more detailed summary of how the EMAC results are being used for $\delta^{13}CO$ might be helpful in this respect.

3) While I agree with significant parts of this paper, the third paragraph of section 3 has several things that I cannot agree with. For example, the Allan *et al* papers did not just consider seasonal cycles in the $CH_4$ data. Their consistency with a total $CH_4$ budget based on other work was inherent throughout those analyses – e.g. see Table 1 of Allan *et al* (2001a). Similarly, early work to extract phase diagrams for variations in $\delta^{13}CH_4$ vs those in $CH_4$ mixing ratio, as shown in Figs 8 and 9 of Allan *et al* (2001a), had explicitly removed trends from the data using the very detailed Seasonal-Trend-Loess (STL) method and so it is not correct to imply that these results would have been sensitive to long term trends.

   Also, the point about having to take account of a reversal in the long-term trend for $\delta^{13}CH_4$ as shown in Nisbet *et al* (2016) will apply to the analysis done in this paper as well. In particular, although not shown in Nisbet *et al* explicitly, that analysis has a reversal in trends for the $CH_4$ source $\delta^{13}C$ occurring around 1994 – 1996 which is also when there is a maximum in $\delta^{13}CO$ in the ETSH. That shows, again, the much faster response of the short-lived CO than the longer-lived $CH_4$. Consequently, concerns about dealing with trends in the $CH_4$ budget can be even more pertinent for this analysis.

4) A broader concern that I have with section 3 is that this is not covering how the EMAC model may differ from other models such as TOMCAT used in Hossaini *et al* (2016) and which produces a much higher estimate for Cl in the marine boundary layer. These estimates will be very dependent on how details such as aerosol transport and DMS chemistry are treated. But comparison of the MESSy AIRSEA submodel used in EMAC with the GLOMAP aerosol microphysics model used in TOMCAT does not seem to have been considered anywhere so far. George Box is often cited as saying "All models are wrong, but some are useful" but the bigger problem with atmospheric chemistry models is that they all tend to hide the details at levels that make it virtually impossible to decide which is actually the useful one. Solving that problem is outside the scope of this paper, but it would be helpful if the issue was raised.

5) The conclusion in section 4 may be the only part of this paper that some will read. On that basis I would argue that it should have a short summary of the range of different estimates for $CH_4$ removal by tropospheric Cl and their basis. E.g. Vogt *et al* (1996) showed that autocatalytic release of halogens from sea salt should be expected and several subsequent publications on aerosol chemistry have made similar points. Allan *et al* (2001b) then used such estimates of Cl concentration to derive an initial estimate for the magnitude of this sink, but that estimate tended to increase in subsequent papers to become as large as 25 ± 12 $TgCH_4$/yr. The more recent Hossaini *et al* (2016) treatment of marine air chemistry derived a tropospheric Cl methane sink of ~12–13 $TgCH_4$/yr and noted that there could be some larger regional effects. Then this paper is reducing the Cl sink again and now even more significantly. The basis for such a reduction and its implications for the $CH_4$ and CO budgets can then be summarised much as is done currently.

In conclusion, I would restate that this paper sets out an important extension of the work done previously on the potential role of $CH_4$ + Cl in explaining the $\delta^{13}CH_4$ data. E.g. while Lassey *et al*, 2011, sets out the sensitivity of an 'apparent KIE' to small variations in the sources, that made no mention of how this might be seen in $\delta^{13}CO$. This paper also sets out a reason why all future analyses

of $\delta^{13}CH_4$ data would ideally include a consistency check with $\delta^{13}CO$, but unfortunately the limited spatial and temporal coverage for $\delta^{13}CO$ data will still prevent that.

At the same time, I do not think that this treatment of the two periods 1994 – 1996 and 1998 – 2000 is conclusive. In particular the much shorter lifetime of CO makes interpretation of its data much more susceptible to interannual changes in the source $\delta^{13}C$, and in the Southern Hemisphere these are expected to be relatively larger than in the Northern Hemisphere. Also, coverage of the seasonal cycles for $\delta^{13}C$ in both $CH_4$ and CO, with their variations from an equilibrium state, are not clear. The significant differences between two recent and detailed atmospheric chemistry model-based estimations of Cl in the MBL also raises other questions.

Some of my concerns may be too deep to be resolved by one paper, but I would like some parts of this one to be improved, and that it then be published in order to move towards a more conclusive understanding of how we should interpret the growing amount of $CH_4$ isotopic data.

**Specific comments**

line(s):
12 – 18: Another point that should be brought into the introductory paragraph is that the growing spatial and temporal coverage in $\delta^{13}CH_4$ data means that they are now being used for top-down estimates of changes in the source – sink budget to the order of ~1%.

30 – 39: I would suggest that this coverage of KIE also mention Barker *et al*, 2012 (references given below) which used an *ab initio* approach in quantum chemistry to determine the KIE for $CH_4$ + Cl. That showed theoretical calculations for $^{12}C/^{13}C$ rate constants are close to experimental estimates but a bit smaller. However, the authors accept that there are still some issues to be resolved with that method.

40 – 46: Somewhere, and probably in this paragraph, the point should be made that, while there is also a CO + Cl removal process, the rate constant for that is typically six times smaller than that for CO + OH, whereas the rate constant for $CH_4$ + Cl is typically 20 times larger than that for $CH_4$ + OH. Therefore, Cl is not expected to play a significant role in tropospheric CO removal, except possibly at polar sunrise (Hewitt *et al*, 1996) and it is included in some stratospheric chemistry analyses, see Sander *et al* (2011).

47 – 55: This is an important point – i.e. that anomalies observed for $\delta^{13}CO$ in both the Antarctic and Arctic are very likely to be caused by stratospheric Cl as shown by Jobson *et al,* 1994, so they do not provide evidence for a wider role due to tropospheric Cl.

62: The Young *et al* reference mentioned here is for a study of the night time urban atmospheric chemistry budget in Los Angeles. So, it is not clear why that might be relevant here.

77: I would suggest that the wording be changed here to avoid this sounding like the work has a foregone conclusion. E.g. it could be "… inferred from $^{13}C$ isotope enrichment in CH4,  is this effect  visible as concurrent isotope depletion in CO?

120: this is a minor point, but the samples classified as 'Scott Base' in this paper were actually collected at Arrival Heights which is about 4 km from Scott Base in a fenced area labelled 'entry by permit only' and reserved for clean air and electromagnetic studies. Some of the NIWA datasets use the abbreviation AHT for this site.

126 – 132: As noted in the general comments, the longer-term records for CO and $\delta^{13}CO$ show a decreasing trend in the CO mixing ratio after 1998 and a more obvious trend to lower $\delta^{13}CO$ values.

But a more significant issue for the analysis done in this paper is the extent to which interannual variability in the CO budget can alter results based on a constant budget.

128: It seems that this should be citing Gromov, 2013, Sect 4.1.1.

150 – 155: This part of the paragraph brings in results from the following sections and so is hard to follow. Also it is noted here that the data errors are too large to dismiss this 'Cl-driven difference' but the conclusion suggests that such a difference can be dismissed. So I would suggest that these points be moved to section 2.4.

151 – 152: Also the numeric value for 'times smaller than the errors in $\Delta$' is missing in the text.

163 – 166: This paragraph is setting out the basis for Table S1 that gives a global average Cl concentration of 261 atoms cm$^{-3}$ and which is five times less than the equivalent value given in Hossaini *et al* (2016). As noted in general comment #4, because the same emissions and precursors are being used here as in Hossaini *et al* (2016), it raises questions about the models and the need for some explanation as to why the estimated MBL Cl concentrations can differ this much.

170 – 172: This point could be made more clearly by noting that the very small seasonal cycle seen in $CH_4$–derived [CO] is largely due to both its production and its removal being proportional to [OH].

181- 193: Equation (1) is written as an approximation and part of the reason for that is that it applies to a theoretical equilibrium between the sources and sinks rather than to the continual seasonal changes in both. As noted in the general comments, this appears to be the reason why the net fractionation effect seen here is a lot larger than that derived in Allan *et al* (2007), using the UK Unified Model. Also, the seasonal cycle for Cl removal used in the Allan *et al* papers puts this at a significant level for only 3 months. Consequently, if there is a way to give approximations for the non-equilibrium effects then that could clarify this analysis.

Table 2: The layout used for this table could be improved to make it clearer by separating the three sections, which each have different column headings. Also, as the ‡ symbol is only used for the last part of the table it could be made clearer by using a subheading mentioning the Allan *et al*, 2007, paper at the top of that section.

212: It would read a bit better if this sentence started with "Finally, …"

223: This reference to using the same seasonal cycle for OH and Cl is not quite correct as Allan *et al* (2007), and its preceding papers, have used a seasonal variation for Cl in the marine boundary layer based on DMS related species in the Southern Hemisphere and that has a much shorter seasonal cycle than OH.

254: "none of which" can be read as meaning none of the analyses mentioned in this paragraph, whereas Nisbet *et al* (2016) did explicitly consider different spatial and seasonal distributions of Cl removal – see Table 1 in that publication.

**References for the comments above that are not included in the paper by S. Gromov *et al*.**

Allan, W., M. R. Manning, K. R. Lassey, D. C. Lowe, and A. J. Gomez (2001a), Modelling the variation of d13C in atmospheric methane: Phase ellipses and the kinetic isotope effect, GBC, 15, 467-481.

Allan, W., D. C. Lowe, and J. M. Cainey (2001b), Active chlorine in the remote marine boundary layer: Modeling anomalous measurements of d13C in methane, GRL, 28, 3239-3242.

Allan, W., D. C. Lowe, A. J. Gomez, H. Struthers, and G. W. Brailsford (2005), Interannual variation of C in tropospheric methane: Implications for a possible atomic chlorine sink in the marine boundary layer   JGR, 110, D11306, doi:11310.11029/12004JD005650.

Barker, J. R., T. L. Nguyen, and J. F. Stanton (2012), Kinetic Isotope Effects for Cl + CH4 = HCl + CH3 Calculated Using ab Initio Semiclassical Transition State Theory, Journal of Physical Chemistry A, 116(24), 6408-6419.

Giglio, L., J. T. Randerson, and G. R. v. d. Werf (2013), Analysis of daily, monthly, and annual burned area using the fourth-generation global fire emissions database (GFED4), Journal of Geophysical Research (Biogeosciences), 118(1), 317-328.

Hewitt, A. D., K. M. Brahan, G. D. Boone, and S. A. Hewitt (1996), Kinetics and mechanism of the Cl + CO reaction in air, International Journal of Chemical Kinetics, 28(10), 763-771.

Manning, M. R. (1999), Characteristic modes of isotopic variations in atmospheric chemistry, GRL, 26(9), 1263-1266.

Prather, M. J. (1994), Lifetimes and eigenstates in atmospheric chemistry, GRL, 21, 801-804.

Randerson, J. T., G. R. van der Werf, G. J. Collatz, L. Giglio, C. J. Still, P. Kasibhatla, J. B. Miller, J. W. C. White, R. S. DeFries, and E. S. Kasischke (2005), Fire emissions from C3 and C4 vegetation and their influence on interannual variability of atmospheric CO2 and $\delta^{13}CO_2$, GBC, 19(2), GB2019.

Sander, S. P., et al. (2011), Chemical Kinetics and Photochemical Data for Use in Atmospheric Studies: Evaluation Number 17, Jet Propulsion Laboratory, Pasadena, California.

Tans, P. P. (1997), A note on isotopic ratios and the global atmospheric methane budget, GBC, 11(1), 77-81.

---

## Referee Comment (RC2) · Anonymous Referee #2 · 14 Apr 2018

The paper presents an interesting new angle on evaluating previous indirect claims of a globally significant role of Cl for the removal of CH4, by analyzing the isotopic composition of the reaction product CO. The results indicate that the previous estimates are strongly overestimated.

The paper is well focused on bringing over a clear message and I appreciate that the authors try to keep it short. However, in particular in section 2 and Fig 2, where the new evidence is presented, the information is very dense, and the authors should help the reader by adding more explanation and making more explicit statements.

The derivation of equations 2 and 3 should be shown explicitly, either in the main paper or in an appendix, and more information / explanation should be added. A parameter lambda_a is introduced, but what is it? An additional parameter mu is introduced, is

this necessary? It replaces the parameter DeltaS that is mentioned in the sentence above Eq. 2 to express the sensitivity, but it does not appear in the equation. This should be motivated better.

Fig 2 shows results obtained with Eq. 2 and 3, but the authors should help the reader by describing this complex figure step by step, linking it to the equations and the data. For example, why is Fig 2 shown as function of alpha, what does this signify, and how does DeltaS enter this figure? What are the units of the numbers in Fig 2a (permil). I cannot fully follow the argumentation of paragraph 25, but it appears important for the paper. The error ranges given in lines 213 ff do not correspond to the error ranges indicated in Fig. 2.

Minor points:

Line 100: It would be useful to spell out precisely what the issue is. It is mentioned indirectly in the following lines (the CH4 derived fraction would be too dominant, line 102), but please provide the line of argumentation explicitly: 1) the bottom-up budget of CO isotopic composition is too negative in 13C compared to observations, 2) the most negative course is CH4, 3) to close the isotope budget required lowering the yield in previous studies, and 4) making CH4-derived CO even more depleted in 13C would aggravate the problem.

Line 165: Can you comment on the difference in Cl levels compared to Hossaini et al. (2016)?

Line 171: On which basis do you "expect" a factor 1/5 lower variation of the CH4-derived CO in the ETSH (I assume compared to the SH)?

Line 176: Provide some more details on eta_C. This is a complicated quantity, and relevant here, so some background should be provided in the paper itself rather than referring to Gromov (2013). Specify next sentence: Is it the difference between the atmospheric isotopic composition and a global averaged source mix or the sources at

this point and space in time?

Line 192: It is not clear what you want to indicate here: "In a statistical sense...." Do you refer to the differences derived for different delta_m values, or the difference between the stations. And further, what does this mean?

Fig 2, and caption. What is the unit of the numbers shown as labels in Fig 2a? Is it useful to show a yield from a personal communication (M. Krol, correct spelling) in the figure without relating it to a reference?

Line 205 ff: Explain better the meaning of the sentence "Importantly, ...". It is not immediately clear that you are less sensitive when you add a sink than when you replace a sink.

Either reword or remove lines 227 – 229. This is a confusing statement.

Line 230 – 232: Motivate where the number of "at least one-third" comes from.

Line 233: Motivate the value of 2 per mill.

Line 234 ff: Phrasing the quantification in terms of lambda values is confusing. We know very well that the yield of CO is higher than 0.12. Is it not more instructive to compare the model results with the experimental data in Fig 2? I.e. discuss the "vertical" offset, which would simply imply less change in the Cl sink, rather than the "horizontal" offset, which projects the real cause of the discrepancy to an unrealistic change in lambda?

Line 235 to 240: Move this paragraph to the description of Figure 2a. Now it is used only in the discussion of the unrealistically low lambda value, but it is helpful for Fig 2a in general.

Line 251/52: Given the timescales for equilibration of the mole fraction and the isotope reservoirs (Tans, P. P.: A note on isotopic ratios and the global atmospheric methane budget, Gl. Biogeochem. Cycles, 11, 77-81, 1997), it seems highly unlikely to me

that there could be a "stabilization" signal in the isotopic composition before the signal occurs in CH4, at least when this is interpreted as a manifestation of steady state between sources and sinks. I suggest replacing "hiatus" by the explicit statement of intermittent stop in the annual growth.

Line 259: …. is useful … For what? And on what basis do you make such a statement? The rest of the paragraph is quite vague, I wonder why you did not investigate the seasonal signal if you suggest that it should be a sensitive indicator for Cl.

Line 277: As mentioned above, I suggest changing the line of argumentation away from the totally unrealistic values for lambda. You can make the point stronger by staying with the possible lambda range.

Line 288: This range of values in the realistic range for lambda should be presented and discussed in the main text, see comments above. I consider this a (the?) main result, which is not well motivated and presented. Also, you should link it to the maximum possible Cl difference between the two periods in the following paragraph, where the message is less clear because the parameter lambda_a is involved again.

Line 289/90: This may be overly optimistic. What about source variations of CO (e.g. bb)? If it is mentioned in the conclusions if should follow from a more detailed discussion in the previous section, but it has not been mentioned before at all.

Line 294: What is lambda_a (see comment above) and why does it come back here in the conclusions, whereas most of the discussion was about lambda?

Line 297: This last sentence is not really about your results, and already known, so although the statement is strong, it does not summarize your analysis. Also, one could argue that this implies that there is still at least one big error in the present understanding of the global CO isotope budget parameter, correction of which could offset the budget in a way that there may be room for Cl again.

Technical points:

[Figure]

Make "incomplete" comparisons complete. As it is now, one has to go back to the previous sentence to exactly locate the reference for the comparison.

Examples:

line 26: It is easier . . . compared to what? (direct measurements)

line 83: . . . are less complicated . . . compared to what? (the NH)

line 171: . . .much lower. . . compared to what?

Line 152: Rephrase sentence "significant or not. . .". If a signal is not significant, don't use it to support a scientific argument. Also, in this sentence you write about effects in both directions, in the next sentence you relate this to CO that would work in a "similar" direction (but here it is only one of them). This is confusing, please clarify.

Caption Fig 1 mentions pluses but they are not visible (probably the small dashes in the boxes where the vertical part of the cross coincides with the vertical line).

Line 172: Reword sentence: "The average fraction of the latter . . .." You write that two values are proportional, but one has a fixed value, so the other one as well.

Line 181: Reword ". . .can be approximated as due to. . ."

Line 211 and line 226: replace cf. ibid. by Fig 2.

Line 224: remove "happens to be", this implicates that this is by chance.

Line 249 and 250: add "period" after HC and LC

Line 291: Why "Nevertheless"? The sentence does not seem to require this logical connection.

---

## Author Comment (AC1) · 8 Jun 2018

**Comments on "A very limited role of tropospheric chlorine as a sink of the greenhouse gas methane" by S. Gromov *et al**

Martin Manning (Referee)

martin.manning@vuw.ac.nz

We are very grateful to Martin Manning for his detailed review and valuable comments. Below we hope to show that it is the putting right that counts.

**General Comments**

Given the number of recent papers that have proposed quite different explanations for the post-2006 rise in atmospheric $CH_4$, it is clear that there are still some major systemic uncertainties in our understanding of the $CH_4$ source – sink budget. While the increasing amount of isotopic ($\delta^{13}CH_4$) data should help to resolve these uncertainties, this has to deal with a limited understanding of removal by Cl with its very large kinetic isotope effect (KIE), and so a significant effect on $\delta^{13}CH_4$ even though most of $CH_4$ removal is by OH.

Analyses by Allan *et al* (2005, 2007) [NB additional references not included in the Gromov *et al* paper are given below] suggested that removal by Cl in the marine boundary layer can match $\delta^{13}CH_4$ data in the Southern Hemisphere, so long as there is a significant amount of interannual variability in the amount of this removal, but the driving factors for such variability in Cl still needed to be clarified. The Lassey *et al* 2011 analysis was then based on a simpler budget approach but showed that small interannual variations in the seasonal cycles for different sources can also lead to an 'apparent KIE' in the data that is quite different to that due to chemical removal processes alone.

More recent work by Hossaini *et al* (2016) has shown that, when a detailed form of tropospheric Cl chemistry is added to the TOMCAT chemical transport model, that sink for $CH_4$ appears to be about half what was used in Allan *et al*, but that there is also the potential for large scale regional effects and higher amounts of Cl in some places.

This new paper by Gromov *et al* is definitely a very important extension to the work cited above, because it now addresses the issue of how a highly fractionating removal of $CH_4$ would affect the atmospheric CO that is produced by both of the $CH_4$ + OH and $CH_4$ + Cl removal processes. A key point made in this paper is that the role of Cl in removal of $CH_4$ must be kept consistent with data and budget analyses for $\delta^{13}CO$, and that the long records of NIWA data in the Southern Hemisphere are very relevant for this. In addition, because $CH_4$ oxidation produces 40 - 50% of the CO that is observed in the Southern Hemisphere, the isotopic effects of removal by Cl should be more evident there.

The EMAC model that is used in this analysis, and the atmospheric chemistry that it covers, are quite well documented in a number of earlier papers and the tagging tools, described in Gromov *et al* 2010, provide a clear way of attributing CO to its different sources. So, the framework used in this paper has a clear basis.

However, there are some aspects of the paper that I find to be either not clear or incomplete as follows.

1) Use of the EMAC model in this work appears to have constant surface sources for CO over the 1994 – 2000 period and so does not include the effects of any trends or interannual source variations. Because the lifetime of CO is about forty times shorter than that for $CH_4$, its mixing ratio and $\delta^{13}CO$ are much more sensitive to interannual variations in its sources. In particular, biomass burning is a significant source of CO in the Southern Hemisphere and its interannual variations are not well known prior to 1996 (e.g. Giglio *et al* 2013). More generally, while burning in C4 ecosystems is known to be dominant, the interannual variations are larger for C3 ecosystems that have a quite different $\delta^{13}C$ (e.g. Randerson *et al* 2005) so there can be relatively larger interannual variability in the source's $\delta^{13}C$ than in its magnitude.

Southern Hemisphere data at the start of the 1998 – 2000 period will also be affected by the extensive biomass burning emissions from Indonesia that continued for a longer period than usual in 1997. This is seen in the NIWA data that have an (admittedly) noisy long-term maximum in CO mixing ratio in 1998 but a much clearer maximum in $\delta^{13}CO$ that year. It is still not clear to what extent these source variations will affect the relationships in the tightly coupled $CH_4$ – CO – OH system, but as noted in Lassey *et al*, 2011, the 'apparent KIE' for $CH_4$ is quite sensitive to variations in seasonal cycles for the sources.

Some structural differences between the two 3-year periods used in this paper are seen quite clearly in Fig S2, e.g. the much smaller amplitude for $\delta^{13}CH_4$ seasonal cycles over 1998 – 2000 as shown in Fig S2(d). Therefore, it would be better to show all of the data this way in the main text, rather than just the statistical summaries currently used in Fig 1. Similarly, Fig S3 is a very clear way of showing model results and would also be useful in the main text.

To summarize this point: given that CO and $\delta^{13}CO$ are much more sensitive to changes in sources than $CH_4$ and $\delta^{13}CH_4$, it is important to consider how the paper's use of a fixed non-$CH_4$ source for CO may have hidden some of the differences between the two periods.

We agree with this summary. Irrespective of the sensitivity of $CH_4$ and $\delta^{13}CH_4$, the use of CO poses upfront a dilemma. On the one hand, we use the sensitivity of $^{13}CO$ to changes in the Cl/OH ratio (thanks to the large KIE form Cl, this gives hope), for which the SH is the best region to test. On the other hand, just here biomass burning forms a major variable source, as pointed out by the Reviewer. The variability in $^{13}CO$ is partly decoupled from that of CO, due to variations in the relative contributions of CO from burning C4 (dominant) or C3 based vegetation. How serious is this effect? Taking the time series of biomass burning source $\delta^{13}C$ for SH (Gromov et al. (2017), Fig. 4 and Table 3), we can put an upper limit of 2‰ for this source $\delta^{13}C$ variation (between 1997 and 2000 averages, $1\sigma = 0.62$‰, see the Figure below). We may only speculate that the variations between in 1994–1996 and 1998−2000 did not exceed this range; on the other hand, about half of these 2‰ is due to the large $\delta^{13}C$ excursion in 1997 triggered by very strong Indonesian fires due to the ENSO pattern in those years.

LONGITUDE : 180E(-180) to 180E
LATITUDE : 90S to 0

DATA SET: EVAL2-isoC-BB-GFED2.1

GFED2.1/ISOLUCP $\delta^{13}C$ of BB-emitted CO in the SH ($^o/_{oo}$ V-PDB)

If one assumes Cl being constant, one can compare the variability of $^{13}CO$ to that of CO to see if changes in biomass burning composition (C3/C4 distribution) have a substantial impact. Referring to Fig. 2, we have calculated the variations in $\delta^{13}C$ of CO and its non-$CH_4$ derived component, which is required to mask the Cl-induced changes. Indeed, it cannot be excluded, that some of the difference between the two periods may have been hidden, or likewise may have been augmented by variations in the contributions of burning of C3 vegetation compared to the total BB source of SH CO, but the effect is demonstrably negligible. We note uncertainties about $\delta_n$ (line 202) which exceed given above variation for BB source $\delta^{13}C$; however even a 4‰ variation $\delta_n$ will add tangible uncertainty to $\Delta\delta_c$ value only at very low $\lambda < 0.25$. We added to the text that the use of a fixed non-methane source has a negligible effect.

We note that we already had stated (Section 2.4, line 425-430) "unless masked by unrealistic concurrent increases in $\delta^{13}CO$ of the non-methane sources of about +(11.6-13.5)‰".

2) It appears from this paper, and from the earlier Gromov (2013), that the EMAC model has produced CO mixing ratio values but not the $\delta^{13}CO$ values directly and that is why there is no analogue for Fig S3 showing model results for $\delta^{13}CO$ as well. Instead, section 2.4 gives formulae that bring together model and observational data, as summarised in Table 2, and then introduce the comparison between high Cl and low Cl periods shown in Fig 2.

Indeed, Gromov (2013) outlined the main problem of mass-balancing $^{13}CO$ for which an extensive modelling study were required. The current work, however, is mostly based on observational data, and we want to use model-derived information as little as possible in order to retain the observational nature of the evidence. Therefore, we use **only** model-derived $\gamma$ and $\eta_c$ values (which are difficult to obtain otherwise) and mass-balance the non-$CH_4$ CO sources' $\delta^{13}C$; for relevant components we show that their uncertainties are typically lower than that in the observed difference in $\delta^{13}C(CO)$ between the HC and LC. Another argument for using the approach via Eq. (1) is that we require annual averages (3-yr QAAs) in order to obtain low enough uncertainties (via larger statistic) about $\delta^{13}C(CO)$ changes.

However, while equation (1) is valid when $CH_4$ and CO are at equilibrium with their average sources and sinks, it does not apply more generally for the seasonal cycles in $CH_4$ and CO mixing ratios and $\delta^{13}Cs$. Neither the mixing ratios nor $\delta^{13}Cs$ are at equilibrium due to the significant seasonal cycles in sources and

removal rates. And, while large differences in lifetimes mean that the dis-equilibrium will be larger for $CH_4$ than for CO, in both cases the isotope ratios are expected be slower at reaching equilibrium than mixing ratios (e.g. Tans, 1997). Furthermore, the tightly coupled $CH_4$ – CO – OH system has different modes of variation (Prather, 1994, etc) and these are different again for the isotope ratios (Manning, 1999).

For the use of Eq. (1) is not relevant that $CH_4$ and its isotopic composition in response to inter- annual variations and short trend changes do not reach equilibrium. The impact of variations in $CH_4$ and $\delta^{13}CH_4$ is small compared to the impact of possible changes in Cl (to a degree as discussed by Allen et al. and in this paper) and of changes caused by a shifting in partitioning between the methane and the non-methane sources.

Concerning the applicability of Eq. (1) to CO and $\delta^{13}CO$, phase ellipses of $CO/\delta^{13}CO$ would be highly adventurous in this sense. Tans, Prather and Manning have indeed pointed out that time constants are longer, especially for isotopic composition, than the commonly used life time or turnover time. If we for argument sake consider the SH OH seasonality to be the sole driving force of SH CO seasonality and we compare the phases of their seasonal cycles, we find a time lag of about 3 months. We compare in our paper however quasi annual averages (QAAs) for CO for the 2 periods of assumed high and low chlorine.

This difference between an equilibrium and dynamic situation appears when comparing the change in $\delta^{13}CH_4$ corresponding to equilibrium conditions for the HC and LC periods, as derived in section 2.4, with results shown in Fig 2b of Allan *et al* (2007), based on runs with the UK Met Office Unified Model, for the same estimates of Cl concentrations. The Allan et al difference in $\delta^{13}CH_4$ between the two different Cl concentrations is less than half that given in section 2.4. This point is mentioned again in specific comments on lines 181 – 193 below.

Therefore, it is not clear to what extent seasonal cycles in the dis-equilibrium and differences in that for both $CH_4$ and CO as well as for mixing ratios and $\delta^{13}C$s, will modify what is summarised in section 2.4. A more detailed summary of how the EMAC results are being used for $\delta^{13}CO$ might be helpful in this respect.

We do not present/regard the difference in $\delta^{13}C(CH_4)$ between the HC and LC periods in Sect. 2.4. Perhaps, Reviewer implied those shown in Sect. 2.2, which do not exceed 0.1‰ (in terms or 3-yr averages in QAAs), which is a negligible change for the $CH_4$-derived $^{13}CO$. The largest effect on the latter are indeed driven by changes in the average $CH_4$ sink fractionation ($\varepsilon_m$) and should be pronounced in $\delta^{13}C(CO)$, however not in $\delta^{13}C$ of $CH_4$ due to its large inventory and potentially varying sources strengths/signatures. The question of equilibrium indeed is key here for CO, whose average lifetime is ~2 months in the troposphere, which means its inventory is reset within one year. We do not regard $CH_4$ at equilibrium here at all (and we do not need to) – therefore we use the QAAs only. Figs. S2 and S3 evidence that there is no disequilibrium in observed/simulated CO. Our approach also does not require de-trending, as compared to phase ellipses method. We are also concerned about Allan et al. (2007) approach using equilibrated $CH_4$ tropospheric inventory in a transport model with composite SST and wind fields, i.e. which likely produce unrealistic $CH_4$ distributions based on atmospheric dynamics and tracer transport/mixing different from those corresponding observational data. To recap: due to much shorter lifetime of CO, our approach is negligibly influenced by the disequilibrium in tropospheric $CH_4$ and its $\delta^{13}C$. However, it is sensitive to rapid changes in $CH_4$ sink $^{13}C$ KIE, which should occur under large Cl variations.

We have amended Sect. 2.3 (also following the comment of Reviewer #2) regarding the use of EMAC results. We also add the seasonal cycles of $\eta_c$ to Fig. S3 in order to facilitate the explanation of this parameter.3) While I agree with significant parts of this paper, the third paragraph of section 3 has several things that I cannot agree with. For example, the Allan *et al* papers did not just consider seasonal cycles in the CH$_4$ data. Their consistency with a total CH$_4$ budget based on other work was inherent throughout those analyses – e.g. see Table 1 of Allan *et al* (2001a). Similarly, early work to extract phase diagrams for variations in $\delta^{13}$CH$_4$ vs those in CH$_4$ mixing ratio, as shown in Figs 8 and 9 of Allan *et al* (2001a), had explicitly removed trends from the data using the very detailed Seasonal-Trend-Loess (STL) method and so it is not correct to imply that these results would have been sensitive to long term trends.

Perhaps, we misinterpreted the statement from Allan et al. (2001a), Sect. 5: "Thus we are averaging over interannual variations and assuming that disequilibrium effects and trends can be taken to be *linear over the period 1993-1996*." However, we note here that assumption on linearity of trends is not applicable (will lead to wrong mixing vs. isotope ration slope or phasing) when actual trend (e.g. Loess component) of $\delta^{13}$C(CH$_4$) reverses earlier than that of CH$_4$.

We change the text of our manuscript to do justice to also the earlier paper by Allan et al. (2001a) which is now included in the literature cited.

Also, the point about having to take account of a reversal in the long-term trend for $\delta^{13}$CH$_4$ as shown in Nisbet *et al* (2016) will apply to the analysis done in this paper as well. In particular, although not shown in Nisbet *et al* explicitly, that analysis has a reversal in trends for the CH$_4$ source $\delta^{13}$C occurring around 1994 – 1996 which is also when there is a maximum in $\delta^{13}$CO in the ETSH. That shows, again, the much faster response of the short-lived CO than the longer-lived CH$_4$. Consequently, concerns about dealing with trends in the CH$_4$ budget can be even more pertinent for this analysis.

We believe that proper dealing with annual cycles, their phases, inter-annual changes and trends of CH$_4$ and $\delta^{13}$CH$_4$ is extremely critical. We have in the case of CH$_4$ a lifetime of a decade and relatively small changes and trends. Tens of papers have dealt with this problem set and uncertainties persist.

For our approach, the changes in CH$_4$ and $\delta^{13}$CH$_4$ are not critical as we consider CO and $\delta^{13}$CO. The fast responses of CO and $\delta^{13}$CO make our analysis much more robust. If Cl had indeed been elevated during the high chlorine period by $19\times10^3$ atoms cm$^{-3}$, the impact on $\delta^{13}$CO would have been fast and detectable.

4) A broader concern that I have with section 3 is that this is not covering how the EMAC model may differ from other models such as TOMCAT used in Hossaini *et al* (2016) and which produces a much higher estimate for Cl in the marine boundary layer. These estimates will be very dependent on how details such as aerosol transport and DMS chemistry are treated. But comparison of the MESSy AIRSEA submodel used in EMAC with the GLOMAP aerosol microphysics model used in TOMCAT does not seem to have been considered anywhere so far. George Box is often cited as saying "All models are wrong, but some are useful" but the bigger problem with atmospheric chemistry models is that they all tend to hide the details at levels that make it virtually impossible to decide which is actually the useful one. Solving that problem is outside the scope of this paper, but it would be helpful if the issue was raised.

In our discussion, section 3, we do not discourse upon how the EMAC model may differ from other models such as TOMCAT used in Hossaini et al (2016) (the latter produces similar [Cl] in similar setup to that of EMAC, but not in the more complex setups which derive much higher local [Cl] in MBL). This is beyond the scope of this paper. We agree with the reviewer that there are shortcomings in the models

and that sources of these shortcomings are hard to identify. The only crucial parameter which adds model-derived uncertainty is the yield value, for which we provide sensitivity.

5) The conclusion in section 4 may be the only part of this paper that some will read. On that basis I would argue that it should have a short summary of the range of different estimates for $CH_4$ removal by tropospheric Cl and their basis. E.g. Vogt *et al* (1996) showed that autocatalytic release of halogens from sea salt should be expected and several subsequent publications on aerosol chemistry have made similar points. Allan *et al* (2001b) then used such estimates of Cl concentration to derive an initial estimate for the magnitude of this sink, but that estimate tended to increase in subsequent papers to become as large as $25 \pm 12$ TgCH$_4$/yr. The more recent Hossaini *et al* (2016) treatment of marine air chemistry derived a tropospheric Cl methane sink of ~12–13 TgCH$_4$/yr and noted that there could be some larger regional effects. Then this paper is reducing the Cl sink again and now even more significantly. The basis for such a reduction and its implications for the $CH_4$ and CO budgets can then be summarised much as is done currently.

Perhaps it is a process akin to humans with their innate hopeful and positive nature doing science. After sheep were identified as a source of methane, estimates peaked. After termites were identified as a source of methane, estimates peaked. After plants were identified as a source of methane, estimates peaked. After methane hydrates were identified, estimates peaked. After tundras were identified, estimates peaked. After atomic chlorine gained attention and was suspected to play a significant removal role in the troposphere, estimates peaked. There may be a gold rush, but not all luster is gold. There is no doubt about the importance of chlorine in chemical /physical process in the troposphere in different environments. Our paper casts, in a largely model independent fashion, very strong doubt on the existing high tropospheric free chlorine estimates.

We note that in two publications, each by many experts on tropospheric methane, the estimated removal of 25 Tg $CH_4$ per year by tropospheric chlorine is listed, based on using the $CH_4$ and $\delta^{13}CH_4$ based estimate. Several studies deal with mechanisms of Cl production, yet global scale estimates are not available, apart from model studies. Much emphasis is on different chemical environments (marine and polluted especially) with little predictive power for the global troposphere. In our paper we cannot make a balance of major studies involved because they mostly deal with different aspects.

In conclusion, I would restate that this paper sets out an important extension of the work done previously on the potential role of $CH_4$ + Cl in explaining the $\delta^{13}CH_4$ data. E.g. while Lassey *et al*, 2011, sets out the sensitivity of an 'apparent KIE' to small variations in the sources, that made no mention of how this might be seen in $\delta^{13}CO$. This paper also sets out a reason why all future analyses of $\delta^{13}CH_4$ data would ideally include a consistency check with $\delta^{13}CO$, but unfortunately the limited spatial and temporal coverage for $\delta^{13}CO$ data will still prevent that.

At the same time, I do not think that this treatment of the two periods 1994 – 1996 and 1998 – 2000 is conclusive. In particular the much shorter lifetime of CO makes interpretation of its data much more susceptible to interannual changes in the source $\delta^{13}C$, and in the Southern Hemisphere these are expected to be relatively larger than in the Northern Hemisphere. Also, coverage of the seasonal cycles for $\delta^{13}C$ in both $CH_4$ and CO, with their variations from an equilibrium state, are not clear. The significant differences between two recent and detailed atmospheric chemistry model-based estimations of Cl in the MBL also raises other questions.

We show absence of a systematic difference in $^{13}CO$ assuming the two periods of high and low chlorine published in the literature. Such large differences in chlorine abundance lasting several years would have affected $\delta^{13}CO$ significantly, especially because of CO its short lifetime. A corresponding "compensation" or "hiding" by changes in the ratio of C3 to C4 biomass burning CO is unrealistic because it would not suffice and one would have to assume a coincidence, because no common mechanism can be identified. We further support this in our preceding study (Gromov et al., 2017). Even more so, for such large variations in chlorine on such a large scale, one still has no explanation.

Some of my concerns may be too deep to be resolved by one paper, but I would like some parts of this one to be improved, and that it then be published in order to move towards a more conclusive understanding of how we should interpret the growing amount of $CH_4$ isotopic data.

We deeply appreciate this thorough review and hope that the changes to the manuscript do justice to this.

**Specific comments**

line(s): 12 – 18: Another point that should be brought into the introductory paragraph is that the growing spatial and temporal coverage in $\delta^{13}CH_4$ data means that they are now being used for top-down estimates of changes in the source – sink budget to the order of ~1%.

Thank you, we do so.

30 – 39: I would suggest that this coverage of KIE also mention Barker *et al*, 2012 (references given below) which used an *ab initio* approach in quantum chemistry to determine the KIE for $CH_4$ + Cl. That showed theoretical calculations for $^{12}C/^{13}C$ rate constants are close to experimental estimates but a bit smaller. However, the authors accept that there are still some issues to be resolved with that method.

Barker et al. conclude in their paper that their KIEs for $^{12}C/^{13}C$ are probably still not satisfactory at the level of theory used. With two existing experimental laboratory studies, results from this model study add little weight.

40 – 46: Somewhere, and probably in this paragraph, the point should be made that, while there is also a CO + Cl removal process, the rate constant for that is typically six times smaller than that for CO + OH, whereas the rate constant for $CH_4$ + Cl is typically 20 times larger than that for $CH_4$ + OH. Therefore, Cl is not expected to play a significant role in tropospheric CO removal, except possibly at polar sunrise (Hewitt *et al*, 1996) and it is included in some stratospheric chemistry analyses, see Sander *et al* (2011).

It is normally not considered. None of a few of papers on tropospheric CO mention chlorine as a sink because of the extremely low abundance of Cl and its negligible reaction rate constant with CO. We are glad about this, because the reaction product is not so nice.

We add a paragraph mentioning that after Par. [6], respectively.

47 – 55: This is an important point – i.e. that anomalies observed for $\delta^{13}CO$ in both the Antarctic and Arctic are very likely to be caused by stratospheric Cl as shown by Jobson *et al,* 1994, so they do not provide evidence for a wider role due to tropospheric Cl.

We are confused by this statement. Jobson used NMHC ratios changes as evidence for chlorine and bromine during polar sunrise, at the surface. They write: "Thus the data from Alert and the ice floe site provide evidence for Cl and Br atom chemistry during the ozone depletion episodes observed at polar sunrise."

62: The Young *et al* reference mentioned here is for a study of the night time urban atmospheric chemistry budget in Los Angeles. So, it is not clear why that might be relevant here.

In this short paragraph of the introduction we list some of the recent work on chlorine chemistry research. When we conclude in our paper that there is little tropospheric chlorine, or less than estimated in some papers, we in no way want to injustice to tropospheric chlorine chemistry research.

77: I would suggest that the wording be changed here to avoid this sounding like the work has a foregone conclusion. E.g. it could be "… inferred from $^{13}$C isotope enrichment in CH$_4$, why is this effect not visible as concurrent isotope depletion in CO?

Thank you, we have changed "has been" to "could be".

120: this is a minor point, but the samples classified as 'Scott Base' in this paper were actually collected at Arrival Heights which is about 4 km from Scott Base in a fenced area labelled 'entry by permit only' and reserved for clean air and electromagnetic studies. Some of the NIWA datasets use the abbreviation AHT for this site.

We have added a respective elucidation.

126 – 132: As noted in the general comments, the longer-term records for CO and $\delta^{13}$CO show a decreasing trend in the CO mixing ratio after 1998 and a more obvious trend to lower $\delta^{13}$CO values.

But a more significant issue for the analysis done in this paper is the extent to which interannual variability in the CO budget can alter results based on a constant budget.

Our analysis (see the figure below) of the NIWA station data between 1990−2006 does not confirm any significant trends in CO (we note that the extreme outliers are removed, as described in the manuscript). That is, for [CO] we obtain −0.10±0.12 [nmol/mol/yr] and −0.3±0.16 [nmol/mol/yr] at BHD and SCB, respectively. For $\delta^{13}$C(CO), the slopes are +0.005±0.016 [‰/yr] and −0.030±0.025 [‰/yr] at BHD and SCB, respectively.

[Figure]

| Equation | y = a + b*x | | |
|---|---|---|---|
| Weight | No Weighting | | |
| Residual Sum of Squares | 50882.55463 | 20885.58591 | |
| Pearson's r | -0.04219 | -0.12608 | |
| Adj. R-Square | -8.12777E-4 | 0.01152 | |
| | | Value | Standard Error |
| CO @BHD | Intercept | 257.69758 | 239.64079 |
| CO @BHD | Slope | -0.09951 | 0.1201 |
| CO @SCB | Intercept | 646.0553 | 311.62369 |
| CO @SCB | Slope | -0.29747 | 0.15604 |

- ●— CO @BHD
- ●— CO @SCB
- — Linear Fit of NIWA CO @BHD
- — 95% Confidence Band of CO @BHD
- — Linear Fit of NIWA CO @SCB
- — 95% Confidence Band of CO @SCB

[Figure]

| Equation | y = a + b*x | | |
|---|---|---|---|
| Weight | No Weighting | | |
| Residual Sum of Squares | 924.55681 | 514.57926 | |
| Pearson's r | 0.01502 | -0.08207 | |
| Adj. R-Square | -0.00236 | 0.00232 | |
| | | Value | Standard Error |
| d13CO @BHD | Intercept | -39.1538 | 32.2358 |
| d13CO @BHD | Slope | 0.00477 | 0.01616 |
| d13CO @SCB | Intercept | 31.00748 | 48.91401 |
| d13CO @SCB | Slope | -0.03026 | 0.02449 |

- ●— d13CO @BHD
- ●— d13CO @SCB
- — Linear Fit of NIWA d13CO @BHD
- — 95% Confidence Band of d13CO @BHD
- — Linear Fit of NIWA d13CO @SCB
- — 95% Confidence Band of d13CO @SCB

Regarding the non-CH$_4$ CO sources interannual variability issue, please see our reply to the general comment above.

128: It seems that this should be citing Gromov, 2013, Sect 4.1.1.

Yes, indeed, thank you for this correction.

150 – 155: This part of the paragraph brings in results from the following sections and so is hard to follow. Also it is noted here that the data errors are too large to dismiss this 'Cl-driven difference' but the conclusion suggests that such a difference can be dismissed. So I would suggest that these points be moved to section 2.4.

We agree to refrain from using the EMAC-derived estimate here. Indeed, it is enough to conservatively assume up to 50% of CO derived from CH$_4$ and project the Cl-driven change using this figure.

151 – 152: Also the numeric value for 'times smaller than the errors in Δ' is missing in the text.

Corrected.

163 – 166: This paragraph is setting out the basis for Table S1 that gives a global average Cl concentration of 261 atoms cm-3 and which is five times less than the equivalent value given in Hossaini *et al* (2016). As noted in general comment #4, because the same emissions and precursors are being used here as in Hossaini *et al* (2016), it raises questions about the models and the need for some explanation as to why the estimated MBL Cl concentrations can differ this much.

We are confused about this statement. Hossaini *et al.* (2016) write (Sect. 3.4): "Figure 9 shows the simulated annual mean surface [Cl]. We find that CH$_3$Cl oxidation provides a small [Cl] background of around **0.5–2×10$^2$ atoms cm$^{-3}$ throughout most of the global boundary layer**. When VSLSs are also considered (i.e., ORG2), annual mean [Cl] reaches a **maximum of 0.5 × 10$^3$ atoms cm$^{-3}$** in *some coastal regions of the NH*." Inspecting Fig. 9 (ibid.), the latter figure is simulated maxima; we do not see such concentrations the SH MBL, however. We remark that the setup of EMAC resembles the ORG2 setup of Hossaini et al. (2016), which we communicate in Sect. 2.3.

170 – 172: This point could be made more clearly by noting that the very small seasonal cycle seen in CH$_4$–derived [CO] is largely due to both its production and its removal being proportional to [OH].

Thank you, we agree, simultaneous sink/production of CH$_4$ and CO via OH is indeed the key factor here. We amend the statement accordingly.

181 – 193: Equation (1) is written as an approximation and part of the reason for that is that it applies to a theoretical equilibrium between the sources and sinks rather than to the continual seasonal changes in both. As noted in the general comments, this appears to be the reason why the net fractionation effect seen here is a lot larger than that derived in Allan *et al* (2007), using the UK Unified Model. Also, the seasonal cycle for Cl removal used in the Allan *et al* papers puts this at a significant level for only 3 months. Consequently, if there is a way to give approximations for the non-equilibrium effects then that could clarify this analysis.

We reiterate that fractionations derived in Allan *et al* papers cannot be compared here – we also do not derive them. Instead we use nominal KIE ($\varepsilon_m$) which should have caused fractionations derived by Allan *et al.* In other words, Allan *et al.* values are what we denote "effective fractionation" $\eta_c$ for CO. For CH$_4$, we use nominal sink fractionation $\varepsilon_m$ (and denote it with different symbol), which shows how much different in $^{13}$C/$^{12}$C ratio the portion of reacted CH$_4$ molecules (that will become CO) differs from the leftover CH$_4$. We are not affected by the disequilibrium issues whilst regarding short-lived CO, not long-lived CH$_4$ (see other comments above).

Table 2: The layout used for this table could be improved to make it clearer by separating the three sections, which each have different column headings. Also, as the ‡ symbol is only used for the last part of the table it could be made clearer by using a subheading mentioning the Allan *et al,* 2007, paper at the top of that section.

We add indent between the sections of this table (it is favourable to use two columns for the variables always distributed in two categories). Thank you for the hint, we add the footnote ref. to the entire subsection.

212: It would read a bit better if this sentence started with "Finally, …"

Changed.

223: This reference to using the same seasonal cycle for OH and Cl is not quite correct as Allan *et al* (2007), and its preceding papers, have used a seasonal variation for Cl in the marine boundary layer based on DMS related species in the Southern Hemisphere and that has a much shorter seasonal cycle than OH.

We have corrected the statement.

254: "none of which" can be read as meaning none of the analyses mentioned in this paragraph, whereas Nisbet *et al* (2016) did explicitly consider different spatial and seasonal distributions of Cl removal – see Table 1 in that publication.

We drop the last part of this sentence.

**References for the comments above that are not included in the paper by S. Gromov *et al*.**

Allan, W., M. R. Manning, K. R. Lassey, D. C. Lowe, and A. J. Gomez (2001a), Modelling the variation of d13C in atmospheric methane: Phase ellipses and the kinetic isotope effect, GBC, 15, 467-481.

Allan, W., D. C. Lowe, and J. M. Cainey (2001b), Active chlorine in the remote marine boundary layer: Modeling anomalous measurements of d13C in methane, GRL, 28, 3239-3242.

Allan, W., D. C. Lowe, A. J. Gomez, H. Struthers, and G. W. Brailsford (2005), Interannual variation of C in tropospheric methane: Implications for a possible atomic chlorine sink in the marine boundary layer JGR, 110, D11306, doi:11310.11029/12004JD005650.

Barker, J. R., T. L. Nguyen, and J. F. Stanton (2012), Kinetic Isotope Effects for Cl + CH4 = HCl + CH3 Calculated Using ab Initio Semiclassical Transition State Theory, Journal of Physical Chemistry A, 116(24), 6408-6419.

Giglio, L., J. T. Randerson, and G. R. v. d. Werf (2013), Analysis of daily, monthly, and annual burned area using the fourth-generation global fire emissions database (GFED4), Journal of Geophysical Research (Biogeosciences), 118(1), 317-328.

Hewitt, A. D., K. M. Brahan, G. D. Boone, and S. A. Hewitt (1996), Kinetics and mechanism of the Cl + CO reaction in air, International Journal of Chemical Kinetics, 28(10), 763-771.

Manning, M. R. (1999), Characteristic modes of isotopic variations in atmospheric chemistry, GRL, 26(9), 1263-1266.

Prather, M. J. (1994), Lifetimes and eigenstates in atmospheric chemistry, GRL, 21, 801-804.

Randerson, J. T., G. R. van der Werf, G. J. Collatz, L. Giglio, C. J. Still, P. Kasibhatla, J. B. Miller, J. W. C. White, R. S. DeFries, and E. S. Kasischke (2005), Fire emissions from C3 and C4 vegetation and their influence on interannual variability of atmospheric CO2 and δ13CO2, GBC, 19(2), GB2019.

Sander, S. P., et al. (2011), Chemical Kinetics and Photochemical Data for Use in Atmospheric Studies: Evaluation Number 17, Jet Propulsion Laboratory, Pasadena, California.

Tans, P. P. (1997), A note on isotopic ratios and the global atmospheric methane budget, GBC, 11(1), 77-81.

---

## Author Comment (AC2) · 8 Jun 2018

We thank the reviewer very much for the detailed review.

The paper presents an interesting new angle on evaluating previous indirect claims of a globally significant role of Cl for the removal of CH4, by analyzing the isotopic composition of the reaction product CO. The results indicate that the previous estimates are strongly overestimated.
The paper is well focused on bringing over a clear message and I appreciate that the authors try to keep it short. However, in particular in section 2 and Fig 2, where the new evidence is presented, the information is very dense, and the authors should help the reader by adding more explanation and making more explicit statements.

The information is dense indeed and we have made changes to the text to explain better.

The derivation of equations 2 and 3 should be shown explicitly, either in the main paper or in an appendix, and more information / explanation should be added.

We added requested derivation and explanation in Appendix A.

A parameter lambda_a is introduced, but what is it?

As stated, it is an assumed value of the yield of CO from $CH_4$, for which one may project our calculations obtained with the diagnosed $\lambda$ value in EMAC.

An additional parameter mu is introduced, is this necessary? It replaces the parameter DeltaS that is mentioned in the sentence above Eq. 2 to express the sensitivity, but it does not appear in the equation. This should be motivated better.

Parameter $\mu$ is introduced mostly for the sake of notation convenience in Eq. (3). Indeed, we should have used $\mu$ instead of $\Delta S/S$ in paragraph [25]. We put this right in the revised version.

Fig 2 shows results obtained with Eq. 2 and 3, but the authors should help the reader by describing this complex figure step by step, linking it to the equations and the data. For example, why is Fig 2 shown as function of alpha, what does this signify, and how does DeltaS enter this figure? What are the units of the numbers in Fig 2a (permil). I cannot fully follow the argumentation of paragraph 25, but it appears important for the paper. The error ranges given in lines 213 ff do not correspond to the error ranges indicated in Fig. 2.

We would like to keep the caption in its concise and enough descriptive style, that is, we believe that the reader can easily refer to manuscript for the formulation of $\Delta\delta_c$ and $\Delta\delta_n$. The latter are shown as a function

of $\gamma$ or $\lambda$ (we assume of these were referred to as "alpha" by the Reviewer) because these are the major unknowns in the issue dealt here. This is introduced and thoroughly explained in the manuscript; adding this information to the caption of Fig. 2 will be rather redundant, whereas moving it from the manuscript body to the caption would disrupt the narrative. $\Delta S$ is one of the basis parameters used in calculations, i.e. it is not varied to obtain sensitivities (read we do not see how it may enter this figure, apart from indirectly defining the slope of the KIE-only sensitivity shown).

We nonetheless acknowledge that the information in Fig. 2 is presented too densely. Therefore, we have split Fig. 2(a) into two panels showing $\Delta\delta_c$ and $\Delta\delta_n$ separately and amended the caption accordingly. The ranges given in paragraph [26] (lines 213 ff) are indeed the ranges of $\Delta\delta_n$ values obtained under different assumptions, hence these are not the error ranges. In the amended version of Fig. 2(a), bottom panel, we show the errors associated with $\Delta\delta_n$.

**Minor points:**

Line 100: It would be useful to spell out precisely what the issue is. It is mentioned indirectly in the following lines (the CH4 derived fraction would be too dominant, line 102), but please provide the line of argumentation explicitly: 1) the bottom-up budget of CO isotopic composition is too negative in 13C compared to observations, 2) the most negative course is CH4, 3) to close the isotope budget required lowering the yield in previous studies, and 4) making CH4-derived CO even more depleted in 13C would aggravate the problem.

We improve the narrative here by writing (line 100) "As Manning et al. have pointed out, budget closure is …" and mention (line 96) "a negative shift..", which makes it easier to couple the logic between these 2 paragraphs.

Line 165: Can you comment on the difference in Cl levels compared to Hossaini et al. (2016)?

We amend this sentence accordingly.

Line 171: On which basis do you "expect" a factor 1/5 lower variation of the CH4-derived CO in the ETSH (I assume compared to the SH)?

Perhaps it is not clearly formulated. We imply that variation in the $CH_4$-derived [CO] is a factor five less compared to that of the total [CO]. This applies not only to ETSH but to entire troposphere (in tropics and NH this difference will be larger) and is also driven by synchronous sink/production of $CH_4$ and CO via OH. We add a respective elucidation.

Line 176: Provide some more details on eta_C. This is a complicated quantity, and relevant here, so some background should be provided in the paper itself rather than referring to Gromov (2013). Specify next sentence: Is it the difference between the atmospheric isotopic composition and a global averaged source mix or the sources at this point and space in time?

$\eta_c$ is neither of the two; rather it singles out the effect of sink fractionation on $\delta^{13}C(CO)$, assuming atmospheric mixing and transport alter the latter near linearly. In other words, if a KIE in CO+OH reaction were absent, the airborne $\delta^{13}C(CO)$ would be lower by $\eta_c$. We add a respective information on how this quantity is obtained.

Line 192: It is not clear what you want to indicate here: "In a statistical sense. . .." Do you refer to the differences derived for different delta_m values, or the difference between the stations. And further, what does this mean?

We are referring to the difference in $\delta_n$ values derived for different stations (as we hypothesise that there are no significant local sources south of 40°S except $CH_4$ oxidation). Subsequently, it means the hypothesis stating that "derived $\delta_n$ values at two stations are different" is rejected at $p$-value of 0.31, hence these two values highly likely refer to the same isotope signature (we assume the Reviewer implied delta_n variable here).

Fig 2, and caption. What is the unit of the numbers shown as labels in Fig 2a? Is it useful to show a yield from a personal communication (M. Krol, correct spelling) in the figure without relating it to a reference?

Thank you, we have changed Fig. 2 (a) accordingly (see the reply above). We also add a respective reference (Hooghiemstra *et al.*, 2011).

Line 205 ff: Explain better the meaning of the sentence "Importantly, . . .". It is not immediately clear that you are less sensitive when you add a sink than when you replace a sink.

We add a comparison of the included sink from A07 with the total tropospheric CH4+Cl sink simulated in EMAC to emphasise the importance here.

Either reword or remove lines 227 – 229. This is a confusing statement.

We reformulate as: "Assuming that $\lambda < 0.7$ or that $\lambda \sim 1$ would be in conflict with basic principles, *i.e.* photochemical kinetics and/or dry and wet removal processes affecting the intermediates of the $CH_4 \rightarrow CO$ chain, or their erroneous implementation in the global atmospheric models."

Line 230 – 232: Motivate where the number of "at least one-third" comes from.

Thank you, we add "times 0.7" to $(\delta_m + \varepsilon_m)$.

Line 233: Motivate the value of 2 per mill.

We add a respective footnote.

Line 234 ff: Phrasing the quantification in terms of lambda values is confusing. We know very well that the yield of CO is higher than 0.12. Is it not more instructive to compare the model results with the experimental data in Fig 2? I.e. discuss the "vertical" offset, which would simply imply less change in the Cl sink, rather than the "horizontal" offset, which projects the real cause of the discrepancy to an unrealistic change in lambda?

It is a matter of presentation here. We do indeed compare the model estimate to the experimental data, i.e. the latter shows that the sink estimated by A07 can only be supported by [13]CO data when $\lambda$ is 0.12 or less (cf. where the point from SCB "meets" the curves for SH "vertically"). In this sense it is also a comparison of the "horizontal" offset; the latter, however, is caused by the $\lambda$ value for which we do not have any observational data at all.

Line 235 to 240: Move this paragraph to the description of Figure 2a. Now it is used only in the discussion of the unrealistically low lambda value, but it is helpful for Fig 2a in general.

We add a short elucidation to Fig. 2(a) caption (including a reference to the discussion section), as there is no room in the caption for the details presented in this paragraph.

Line 251/52: Given the timescales for equilibration of the mole fraction and the isotope reservoirs (Tans, P. P.: A note on isotopic ratios and the global atmospheric methane budget, Gl. Biogeochem. Cycles, 11, 77-81, 1997), it seems highly unlikely to me that there could be a "stabilization" signal in the isotopic composition before the signal occurs in CH4, at least when this is interpreted as a manifestation of steady state between sources and sinks. I suggest replacing "hiatus" by the explicit statement of intermittent stop in the annual growth.

Thank you, we will adopt the kind suggestion of the Reviewer. Indeed, using "hiatus" would not be appropriate here as it may imply "equilibration", which is not the case. In this sense, the findings of the recognised study by Tans et al. (1997) regarding *equilibration* times of mixing and isotope ratios of $CH_4$ also may not be applicable here. Slower repartitioning of $^{12}CH_4$ source fluxes (as compared to that of $^{13}CH_4$, i.e. change in source signatures) may cause a slightly earlier detectable signal in $\delta^{13}C(CH_4)$; we do not speak of any equilibration here, it is rather a beginning of an equilibration towards a new state.

Line 259: . . .. is useful . . . For what? And on what basis do you make such a statement? The rest of the paragraph is quite vague, I wonder why you did not investigate the seasonal signal if you suggest that it should be a sensitive indicator for Cl.

We reformulate as: "Because oxidation of $CH_4$ is a main source of CO in the ETSH, and the isotopic composition of atmospheric $CH_4$ is better known than that of its sources, it may well be that variation in the annual average value of $\delta^{13}C(CO)$ is more useful variable for estimating [Cl]. The relatively long lifetime and small seasonality in sources result in weak seasonal cycles of mixing ratio and $\delta^{13}C$ in $CH_4$. In contrast, the seasonal cycle of $\delta^{13}C(CO)$ is dominated ..."
Our early attempts to use the seasonal $\delta^{13}C(CO)$ variations indicated a lack of observational data (large uncertainties due to insufficient statistic) for estimating the Cl input signal.

Line 277: As mentioned above, I suggest changing the line of argumentation away from the totally unrealistic values for lambda. You can make the point stronger by staying with the possible lambda range.

Perhaps it is a matter of our writing/presentation style. We write that agreement is only possible if unrealistic yields were to be applied. Because the yield of CO from $CH_4$ is still an issue, we underscore this problem. When we assume given yields to be "true" we enter this discussion, which we do not wish to do here. In the last sentence of the paper (see reviewer's comment below), we reiterate this issue. It is a real problem.

Line 288: This range of values in the realistic range for lambda should be presented and discussed in the main text, see comments above. I consider this a (the?) main result, which is not well motivated and presented. Also, you should link it to the maximum possible Cl difference between the two periods in the following paragraph, where the message is less clear because the parameter lambda_a is involved again.

We cannot consider the range of probable $\lambda$ values as a main result here, as this is not the property we focus on/research in this study, as opposed to the Cl based input to $\delta^{13}C(CO)$. We merely review the range of conferred values of $\lambda$, and conjecture which range is most realistic. Therefore we present the [Cl] estimate using arbitrary $\lambda_a$ values to specifically draw the Reader's attention to uncertainty about CO yield from $CH_4$ (and giving a possibility to derive a better [Cl] estimate through improved estimates of $\lambda$). We

note that in the abstract we quote the [Cl] variations which correspond to $\lambda_a=0.93$, i.e. that derived with EMAC.

Line 289/90: This may be overly optimistic. What about source variations of CO (e.g. bb)? If it is mentioned in the conclusions if should follow from a more detailed discussion in the previous section, but it has not been mentioned before at all.

The resulting small variability in the CH$_4$-derived [CO] and $\delta^{13}$C is exactly the reason why large variations in other CO sources may help to single out the CH$_4$ input differentially. We add an elucidation here.

Line 294: What is lambda_a (see comment above) and why does it come back here in the conclusions, whereas most of the discussion was about lambda?

See the answer to the comment on Line 288 above.

Line 297: This last sentence is not really about your results, and already known, so although the statement is strong, it does not summarize your analysis. Also, one could argue that this implies that there is still at least one big error in the present understanding of the global CO isotope budget parameter, correction of which could offset the budget in a way that there may be room for Cl again.

We agree, however we would like to keep the this sentence to draw reader's attention to the $^{13}$CO budget closure problem (which is linked to the issue regarded here) in the Conclusions section. We therefore reformulate as a less strong statement: "Regarding ..., it is unlikely that tropospheric Cl is as high as assumed in the literature."

**Technical points:**

Make "incomplete" comparisons complete. As it is now, one has to go back to the previous sentence to exactly locate the reference for the comparison.
Examples:
line 26: It is easier . . . compared to what? (direct measurements)

We reformulate this sentence as: "Not only are indirect measurements easier, the use of trace gases that react with OH and Cl also has the advantage that space- and time-averaged values are obtainable."

line 83: . . . are less complicated . . . compared to what? (the NH)

Yes, thank you, we add the NH here.

line 171: . . .much lower. . . compared to what? NH !

See the answer to the minor comment to Line 171 above.

Line 152: Rephrase sentence "significant or not. . .". If a signal is not significant, don't use it to support a scientific argument. Also, in this sentence you write about effects in both directions, in the next sentence you relate this to CO that would work in a "similar" direction (but here it is only one of them). This is confusing, please clarify.

We rephrase the first sentence. Regarding the following one, we see no confusion here: one statement indicates that $^{13}$C-depleted CO is added or removed from the atmospheric reservoir (opposite directions are explained in the parentheses, no "similar" direction is mentioned?). "Similar fashion" mentioned in the next sentence reiterates that changes in sink KIE also add or remove $^{13}$CO from atmospheric reservoir.

Caption Fig 1 mentions pluses but they are not visible (probably the small dashes in the boxes where the vertical part of the cross coincides with the vertical line).

Thank you, indeed. We have replaced pluses with circles (also in the Supplement, Fig. S2).

Line 172: Reword sentence: "The average fraction of the latter . . .." You write that two values are proportional, but one has a fixed value, so the other one as well.

We removed "average".

Line 181: Reword "… can be approximated as due to …"

Done.

Line 211 and line 226: replace cf. ibid. by Fig 2.

Done.

Line 224: remove "happens to be", this implicates that this is by chance.

Reformulated.

Line 249 and 250: add "period" after HC and LC

Done.

Line 291: Why "Nevertheless"? The sentence does not seem to require this logical connection.

Agreed, this is an overlooked leftover from an earlier edit.

**References**

Hooghiemstra, P. B., Krol, M. C., Meirink, J. F., Bergamaschi, P., van der Werf, G. R., Novelli, P. C., Aben, I., and Röckmann, T.: Optimizing global CO emission estimates using a four-dimensional variational data assimilation system and surface network observations, *Atmos. Chem. Phys.*, **11**, 4705−4723, doi: 10.5194/acp-11-4705-2011, https://www.atmos-chem-phys.net/11/4705/2011/, 2011.

---

## Author Comment (AC3) · 8 Jun 2018

[revised manuscript text omitted]

**Domain abbreviations**

| | | | | | |
|---|---|---|---|---|---|
| zonal: | GLOB | Globe (90°S–90°N) | NH/SH | Northern/Southern Hemisphere | |
| | IT/ET | Intra/Extra-Tropics (separated at 23.4°N/°S) | AR/AN | Arctic/Antarctic (above 66°N/°S) | |
| vertical: | SRF/TP | Surface (lowest model layer) | T | Troposphere (below the TP) | |
| | (M)BL | (Marine) Boundary Layer | FT | Free Troposphere (above the BL, below the TP) | |
| | TP | Tropopause | LMS | Lowermost Stratosphere (above the TP) | |

---

## Author Response (AR1)

**Letter to the Editor on the ACP manuscript "A very limited role of tropospheric chlorine as a sink of the greenhouse gas methane" by S. Gromov et al.**

S. Gromov, on behalf of all authors
sergey.gromov@mpic.de

Dear Editor,

With this letter we would like to summarise the changes and updates to the manuscript we have introduced in the revised version of our paper. We have considered all comments and accepted the majority of the suggestions proposed by the Referees, except a few instances that either were unclear to us or for which we have provided alternative formulation/argumentation.

The structure of the manuscript underwent minor modifications. Following reviewer suggestions, we have introduced Appendix A detailing the derivation of the formulae used and moved the first paragraph of the Conclusions section to the very end, which we believe is the most important part of the message we would like to convey.

The content of the manuscript was amended, as shown in the marked-up version of the manuscript (Author comment AC3). Figs. 1 and S1 underwent minor improvement w.r.t. the whisker plots; Fig. 2 was amended substantially. Please note that two more references were added.

Finally, our final Author Comments (AC1 and AC2) enumerate all referee comments and suggestions and the measures we have taken to address them. We appreciate very much the time you spent for editing this paper.

With best regards,

Sergey Gromov